



# Assessing the impacts of agricultural managements on soil carbon stocks, nitrogen loss and crop production — a modelling study in Eastern Africa

Jianyong Ma[1], Sam S. Rabin[1,2], Peter Anthoni[1], Anita D. Bayer[1], Sylvia S. Nyawira[3], Stefan Olin[4], Longlong Xia[1], Almut Arneth[1,5]

[1] Institute of Meteorology and Climate Research-Atmospheric Environmental Research, Karlsruhe Institute of Technology, 82467 Garmisch-Partenkirchen, Germany

[2] Department of Environmental Sciences, Rutgers University, New Brunswick, 08901 NJ, USA

[3] International Center for Tropical Agriculture (CIAT), ICIPE Duduville Campus, P O Box 823-00621 Nairobi, Kenya

[4] Department of Physical Geography and Ecosystems Science, Lund University, 22362 Lund, Sweden

[5] Institute of Geography and Geoecology, Karlsruhe Institute of Technology, 76131 Karlsruhe, Germany

*Correspondence to*: Jianyong Ma (Jianyong.ma@kit.edu)

**Abstract** Improved agricultural management plays a vital role in protecting soils from degradation in Eastern Africa. Changing practices such as reducing tillage, fertilizer use or cover crops are expected to enhance soil organic carbon (SOC) storage, with climate change mitigation co-benefits, while increasing crop production. However, the quantification of cropland managements' effects on agricultural ecosystems remains inadequate in this region. Here, we explored seven management practices and their potential effects on soil carbon (C) pools, nitrogen (N) losses, and crop yields under different climate scenarios, using the dynamic vegetation model LPJ-GUESS. The model performance is evaluated against observations from two long-term maize field trials in western Kenya and reported estimates from published sources. LPJ-GUESS generally produces soil C stocks and maize productivity comparable with measurements, and mostly captures the SOC decline under some management practices that is observed in the field experiments. We found that for large parts of Kenya and Ethiopia, an integrated conservation agriculture practice (no-tillage, residue and manure application, and cover crops) increases SOC levels in the long term (+11% on average), accompanied by increased crop yields (+22%) in comparison to the standard management. Planting nitrogen-fixing cover crops in our simulations is also identified as a promising individual practice in Eastern Africa to increase soil C storage (+4%) and crop production (+18%), with low environmental cost of N losses (+24%). These management impacts are also sustained in simulations of three future climate pathways. This study highlights the possibilities of conservation agriculture when targeting long-term environmental sustainability and food security in crop ecosystems, particularly for those with poor soil conditions in tropical climates.

## 1 Introduction

Soils contain the largest amount of organic carbon (C) in terrestrial ecosystems, storing around 1500 petagrams of carbon (Pg C) globally (Lal, 2004). However, substantial losses of soil organic carbon (SOC) have occurred over the last decades, arising from agricultural intensification and the continuous conversion of natural soils for agricultural uses to support the food demand of a growing population (Olsson et al., 2019). The estimates of cumulative SOC loss from agricultural land vary widely, with a range of 30 to 160 Pg C across the globe for the post-1850 period (Ruddiman, 2003; Lal, 2004; Pugh et al., 2015; Smith et al., 2016; Sanderman et al., 2017). This soil carbon loss contributes to greenhouse gas (GHG) emissions into the atmosphere and thus accelerates global warming. Increasing SOC storage in agricultural ecosystems through improved management practices has been repeatedly discussed as a promising option to mitigate climate change (Smith et al., 2020; Arneth et al., 2021), with co-benefits for soil fertility and crop production (Seufert et al., 2012; Knapp and van der Heijden, 2018; Shang et al., 2021).



Conservation Agriculture (CA)—particularly the use of minimum soil disturbance (e.g., zero tillage), organic matter addition (e.g., crop residue retention and cover crops), and species diversification through crop rotation—is the most well-known practice to potentially enhance SOC sequestration and improve agricultural sustainability in the sub-Saharan Africa (SSA; Thierfelder et al., 2013; Smith et al., 2016). Much experimental evidence has indicated that SOC stocks under CA systems are significantly higher than conventional farming practices in well-managed trials for SSA (Pittelkow et al., 2015a; Cheesman et al., 2016; Powlson et al., 2016; Sommer et al., 2018) but vary across the region due to the differences in soil properties, climate condition, and the specific management implemented in farming systems (Sanderman et al., 2017). It has been estimated that CA would increase cropland SOC by 1.2 to 2.4 Pg C over both the Eastern and Southern African regions if soil-conserving techniques were completely implemented over 20 years (Zomer et al., 2017). However, adoption of CA remains a challenge in the region as positive crop production effects over time can be hidden by large interannual variability in yields (Giller et al., 2009; Corbeels et al., 2014; Stevenson et al., 2014; Pittelkow et al., 2015b). Furthermore, nitrogen (N) trace gas emissions and nitrate leaching related to agricultural fertilizer also need to be investigated; these N-associated losses from agriculture have negative effects on air quality, freshwater systems, and climate from regional to global scales (Reay et al., 2012; Olin et al., 2015a; Tian et al., 2020).

Process-based ecological models with soil carbon-nitrogen (C-N) dynamics have the potential to understand and quantify the trade-offs between yields, carbon sequestration, and negative environmental effects on larger spatial scales and longer temporal perspectives due to their mathematical representation of plant growth, organic matter input, and soil decomposition (Parton et al., 1993; Li et al., 1994). These models have been widely used to explore SOC response to alternative management practices in different cropping systems (e.g., Century, Lugato et al., 2015; LPJ-GUESS, Olin et al., 2015a; RothC, Mesfin et al., 2021; LPJmL, Porwollik et al., 2021). However, compared to temperate crop ecosystems—particularly in heavily-studied North America, Western Europe, and East Asia—there are limited SOC modelling studies in the tropical agro-ecosystems of SSA (Lemma et al., 2021; Nyawira et al., 2021). This may partially reflect the relative paucity of long-term and field-based SOC measurements in the region (Powlson et al., 2016), which limits the calibration and implementation of process-based models in assessing the impacts of land managements on SOC dynamics in the tropical SSA. In one example, Nyawira et al. (2021) examined DayCent model performance in simulating SOC and crop yields' responses to improved management practices (e.g., manure and crop residue application) for maize-based cropping systems, using experimental data from two long-term field sites in western Kenya. Although model results showed a fairly good agreement with observations, the authors suggested that future model evaluation for other managements (such as cover crops) in sequestering SOC and/or reducing N losses through leaching and gaseous N emissions would be needed to support the recommendation of sustainable agricultural practices in the tropics of SSA. To date, no studies have applied process-based models on the regional scale to detect the long-term joint impacts of environmental change and alternative management practices on associated changes in crop production, C sequestration, and cropland N losses in Eastern Africa, a region where agricultural soils have been experiencing strong degradation due to the combined effects of agricultural intensification and mismanagement over recent decades (Wynants et al., 2019; Mugizi and Matsumoto, 2020).

Thus, in this study, accounting for the most common and important soil-conserving techniques implemented by smallholder farms (such as conservation tillage, mineral fertilizer, and organic matter/manure incorporation), we employ a process-based dynamic vegetation model (LPJ-GUESS, Smith et al., 2014; Olin et al., 2015b) to explore and quantify the effectiveness of these alternative management practices that aim to enhance soil carbon and/or mitigate the negative effects of agriculture on the N cycle. Model results are extensively tested on experimental data from long-term (>10 years) field trials in western Kenya and compared against country-level yield statistics, as well as region-level cropland SOC stocks from published sources. The management effects on soil C pools, crop yields, and N losses in Kenya and Ethiopia are subsequently investigated under present and future climate scenarios. The model-based and large-scale quantification of these management impacts on crop ecosystems provides a scientific understanding for





identifying strategies that possibly minimize environmental negative effects while still addressing society's growing needs for food production, allowing recommendations for sustainable agricultural practices under different farming systems in the tropics of SSA.

## 2 Methods

### 2.1 Model description

LPJ-GUESS is a dynamic vegetation model with process-based representation of plant physiological and biogeochemical processes designed for regional to global applications (Smith et al., 2014). The model has been widely used to investigate vegetation and soil C-N dynamics and their interactions in response to both environmental changes and management, such as changes in climate, atmospheric $CO_2$ concentration, N input (deposition and fertilizer rates), or irrigation. Three distinct land-use types are represented in the model: natural land, pasture, and agricultural land. Vegetation on natural land is described by the growth, disturbance, and mortality of 12 plant function types (PFTs), which differ in their bioclimatic preferences, morphological traits, and growth strategies. $C_3$ and $C_4$ grasses are modelled to represent pastures, with 50% of above-ground biomass removed each year at harvest; the rest, together with root biomass, is assumed to return to soils as litter (Lindeskog et al., 2013). Croplands in the model are characterized by four crop functional types (CFTs; i.e., two temperate $C_3$ crops sown in spring and autumn, a $C_4$ crop representing maize, and a tropical $C_3$ crop representing rice), with crop-specific processes including C-N allocation, plant development stages, and explicit sowing and harvest representation at daily temporal resolution (Olin et al., 2015b). Crops in LPJ-GUESS are prescribed as either rain-fed or irrigated, with their proportions given as an external input (Lindeskog et al., 2013). Recent relevant developments include the implementation of soil N transformation (Olin et al., in prep) and two new legume CFTs (i.e., soybean and pulses) with symbiotic biological N fixation (BNF; Ma et al., 2021).

Soil C-N dynamics in LPJ-GUESS are simulated by a soil organic matter (SOM) scheme derived from the Century model (Parton et al., 1993), in which SOM and litter are characterized by 11 pools with prescribed C:N ratios and decay rates (Smith et al., 2014). The transfer of SOM between pools drives N mineralization or immobilization, as a result of the altered C:N ratios in the donor and receiver pool. Soil mineral N after the process of mineralization and immobilization is partially depleted by plant N uptake, which is assumed to be proportional to plant root biomass and is constrained by soil temperature, plant N status, and the mineral N pool itself (Zaehle and Friend, 2010; Warlind et al., 2014). Leaching of mineral N is a function of the remaining nitrate concentration, percolation rate and available soil water content. N losses through organic leaching is also included in LPJ-GUESS and associated with soil sand fraction, percolation and the size of soil microbial SOM N pool (Smith et al., 2014; Warlind et al., 2014). Gaseous N emission produced in the soil to the atmosphere is simulated as $NH_3$, $NO$, $N_2O$ and $N_2$, with the representation of soil N dynamic processes including ammonification, nitrification, and denitrification in the SOM pools. In this study, we combine N leaching and N gases emission into one value to represent total N loss from crop ecosystems. The model schematic and other calculations on cropland C-N cycles follow an earlier version of LPJ-GUESS described in Smith et al. (2014) and Olin et al. (2015b).

### 2.2 Alternative management practices

Agricultural management options incorporated in the model include variable sowing and harvest dates, irrigation, cover crop grass between two growing seasons, crop residue management, N fertilizer application, and tillage. The latter four practices are varied in the evaluation of management options in this study and described in detail below.

#### 2.2.1 Cover crops

Using cover crops as 'green manure' in-between the main cropping seasons is an effective practice to build up or maintain soil fertility, as they can enrich soil N and soil organic carbon contents if their biomass is fully tilled into the soil. Cover crops implemented in LPJ-





GUESS are modelled as $C_3$ and $C_4$ grasses grown between two agricultural growing periods of main crops, if the fallow duration

exceeds 15 days. The cover-crop leaf and root biomass are added to the surface and the soil metabolic/structural SOM pools, respectively, at the sowing date of the subsequent main crop. N-fixing herbaceous legumes such as Kenya white clover (*Trifolium johnstonii* Oliv.) and Alfalfa (*Medicago sativa*) are sometimes rotated or used as intercrops between cereals to improve the soil quality in East African smallholder farming systems (Sileshi et al., 2008; Muoni et al., 2019). We thus incorporate the process of biological N fixation to $C_3$ grass in the model, following Liu et al. (2011), to account for the effects of herbaceous legumes on C-N cycles in crop

ecosystems. The evaluation of soil C stocks, N leaching, and crop production in response to different cover crop types will be published in a forthcoming paper, and therefore is not presented and discussed here. Grain legumes as cover crops are not yet implemented in LPJ-GUESS.

### 2.2.2 Residue retention

Leaving crop residues in field after harvest can prevent soil degradation, with this also retaining water and nutrients (Smith et al.,

2012). In the standard LPJ-GUESS set-up, this practice is represented by removing 75% of the above-ground biomass after harvest, thus returning the remaining 25% of C and N mass to the soil litter pool for decomposition. In this study, we increase the residue removal fraction to 90% in the model experiments based on the investigated data in Ethiopia (Laekemariam et al., 2016; Lemma et al., 2021), where most smallholder farms practice mixed crop-livestock systems in which crop residues are usually removed from fields after harvest and used as fodder for livestock (Valbuena et al., 2012; Baudron et al., 2014).

### 2.2.3 N fertilizer and manure application

Application of N fertilizer in agricultural land is an important and widespread practice in improving crop production and enhancing SOC storage. However, if not managed appropriately, this practice can easily give rise to negative environmental impacts, like increasing soil $N_2O$ emission (Reay et al., 2012) and/or promoting nitrate leaching to waterways (Tian et al., 2020). N fertilizer in LPJ-GUESS is applied as forms of mineral N and manure. Synthetic fertilizer application takes place at three crop development

stages—sowing, halfway through the vegetative phase, and flowering—with different application rates depending on crop type (Olin et al., 2015b; Ma et al., 2021). All manure is applied to crops at the time of sowing as a single application to reflect real-world practices that account for the time required for manure N to be made available to plants. The manure application in the model is represented as N addition to metabolic and structural SOM N pools with the equal application rate. The amount of C added to soils via manure is then computed assuming a prescribed C:N ratio. The default manure C:N value of 30 (Olin et al., 2015a) was chosen to

represent the C and N content from sources ranging from poultry waste (C:N of ca. 15) to straw-rich manure from livestock (C:N of 40 or more). Here, we adjust the C:N ratio of farmyard manure to 16 in all the model experiments, following the literature-based value for smallholder farming systems in Eastern Africa (Gichangi et al., 2006; Nyawira et al., 2021).

### 2.2.4 Tillage

Different forms of tillage have been used to increase the release of nutrients from the soils for uptake by crops, but the mechanical

disturbance of the soil profile increases soil erosion and heterotrophic respiration and thus enhances soil C losses to the atmosphere (Chatskikh et al., 2009; Badagliacca et al., 2018). Tillage is implemented in the model using a tillage factor, which accelerates the soil decomposition on agricultural land in the surface microbial and humus SOM C pools, and the microbial and slow turnover C pools of the soil. To account for the long-term effects on heterotrophic respiration (Pugh et al., 2015; Olin et al., 2015a), the tillage factor is assumed to be a fixed value of 1.94, which is taken from Chatskikh et al. (2009) and used to modify the decay rate of the four SOM

pools throughout the year.





### 2.3 Experimental set-ups

Our study is divided into three parts. In the first part we examine the model's ability to simulate the SOC and maize yield response to various managements by comparing with observed data from two long-term field sites in Kenya. Next, we update the growth parameters for sorghum in the model to better represent the agricultural production in Eastern Africa because of this widely grown
crop in the region. Yields for six crop types, including the new sorghum parameterization (see Sect. 2.3.2 below), are evaluated against FAO-based statistics in Kenya and Ethiopia. In the last part, the isolated effects of each alternative management practice are first investigated for the historical period and subsequently explored under future climate scenarios by forcing the model with simulated climate over the 21st century from five general circulation models (GCMs, Eyring et al., 2016).

In order to build up cropland soil C and N pools, all simulations were initialized with a 500-year spin-up using atmospheric $CO_2$ from
1901 and repeated de-trended 1901-1930 climate (see Table 1 for data information). During spin-up, potential natural vegetation (PNV) was simulated for the first 470 years, and then the cropland fraction linearly increased from zero to the first historic value (1901) during the last 30 years of spin-up. Model input data is summarized in Table 1, with the different experiment set-ups explained in detail below.

### 2.3.1 Model evaluation at site scale

To evaluate the model performance, we use data from two long-term experimental sites (INM3 and CT1) managed by the International Center for Tropical Agriculture (CIAT) since 2003. The INM3 trial (34.40°E, 0.14°N) is designed to study soil fertility effects of manure and maize residue retention under conventional tillage systems, while the CT1 trial (34.41°E, 0.13°N) mainly evaluates the combined effects of conservation tillage and residue application on SOC dynamics in maize-based cropping systems (Sommer et al., 2018). A total of 16 different trials for the period 2003-2015 in a continuous maize system (two cropping periods a
year) at the INM3 site were simulated: 0 and 4 t ha$^{-1}$ (dry matter) of manure application with 2 t ha$^{-1}$ maize residue retention or removal under four treatments of mineral N fertilizer addition (0, 30, 60, and 90 kg N ha$^{-1}$). Similar simulations over the same period were performed at the CT1 site, but with the difference that minimum and conventional tillage are dominant practices and no trials receive any manure application. At present, within-year double-cropping has not yet been implemented in LPJ-GUESS (Olin et al., 2015a) since the second, "short rainy" growing season is—from a yield perspective—not hugely relevant for most regions of Eastern
Africa (Wainwright et al., 2019). In this study, the second growing period under the continuous maize systems was modelled as a non-N-fixing cover crop with all the above-ground biomass removed from the field at both sites. To parameterize the N application and residue retention practices in the model, the application rate of 4 t ha$^{-1}$ of manure dry matter was converted to 70 kg N ha$^{-1}$ by assuming an N content of 1.75% in farmyard manure with a fixed C:N ratio of 16 (Gichangi et al., 2006). The residue management with 2 t ha$^{-1}$ retention was set to 50% of maize residue left in the field, following the reported proportion described in Sommer et al.
(2018) and Nyawira et al. (2021). In addition, we switched off (on) the tillage option in the model to represent the minimum (conventional) tillage experiment at CT1. A summary of these trials is available in Table 2.

The gridded daily climate data set from GSWP3-W5E5 (Dirmeyer et al., 2006; Lange, 2019; Cucchi et al., 2020) at 0.5° resolution was used, the grid cell with coordinates 34.25°E and 0.25°N is representative for the two experimental sites. We compared the required input variables from GSWP3-W5E5 with site-based weather observations, finding that the gridded climate data had a fairly
good agreement with field records, although precipitation diverged between two data sets on individual days over the experimental period (Fig. S1, in the Supplement). There was not much information available on the land use in years prior to the field experiments. Therefore, to maintain soil N and C pools in equilibrium after model spin-up, we followed the simulation set-ups in Nyawira et al. (2021) and assumed that INM3 was under grassland systems for the period 1901-2002 (A1, Table 1), while at CT1 grassland was





simulated from 1901 to 1991. After this the land use for CT1 trials was implemented according to information provided in the literature (Sommer et al., 2018): rain -fed maize farming systems from 1992-1994 (unfertilized), followed by a crop-fallow period of 1995-2000 (grassland), then two years with fertilized maize (18 kg N ha$^{-1}$) until 2002 (A2, Table 1). In addition, soil physical properties in the topsoil at both sites, such as clay content (%) and bulk density (g cm$^{-3}$), were taken from Sommer et al. (2018) and used as external inputs to further calculate corresponding soil hydraulic properties in LPJ-GUESS (Olin et al., 2015a).

Model performance was assessed comparing the simulated and observed maize yields and SOC stocks in response to varying management practices. For SOC comparison, the measured SOC values were scaled to 0-150 cm from the original depth (0-15 cm) to match the modelled soil depth, using the empirical depth distribution functions proposed by Jobbágy and Jackson (2000):

$$Y = 1 - \beta^d \tag{1}$$

$$SOC_{150} = \frac{1 - \beta^{150}}{1 - \beta^{15}} \times SOC_{15} \tag{2}$$

where $Y$ is the cumulative proportion of the SOC pool from the surface to depth $d$ (cm) and $\beta$ is the relative rate of decrease in SOC stock with depth depending on the measured SOC content along the soil profile. The value of $\beta$ is obtained from the existing literature and set as 0.971 for INM3 and 0.974 for CT1 (Nyawira et al., 2021); $SOC_{15}$ and $SOC_{150}$ represent the cumulative SOC stock at 0-15 cm and 0-150 cm, respectively.

### 2.3.2 Regional crop yields evaluation

In this study we performed simulations with six CFTs—maize, pulses, sorghum, wheat, rice, and soybean—which are grown widely in Kenya and Ethiopia (FAOSTAT, 2021). In a previous modelling study (Olin et al., 2015a), sorghum in LPJ-GUESS was simulated as the maize CFT. Here, we developed a CFT that better represents assimilate allocation to sorghum organs based on the data from Penning de Vries et al. (1989) (Fig. S2, Table S1). The performance of the model for sorghum and five other crops were evaluated by comparing the simulated and reported yields at country level. For regional comparison, the crop yield statistics were collected from FAOSTAT (2021) while the simulated gridded yield (B1, Table 1) was aggregated to national level using land-use maps (described below):

$$Yield_{country} = \frac{\sum_{i=1}^{n} [(Yield_{rain})_i \times (Area_{rain})_i + (Yield_{irri})_i \times (Area_{irri})_i]}{\sum_{i=1}^{n} [(Area_{rain})_i + (Area_{irri})_i]} \tag{3}$$

where $Yield_{country}$ is the aggregated yield in Kenya or Ethiopia; $i$ is the number of grid cells in that country, varying from 1 to n; $Yield_{rain}$ and $Yield_{irri}$ denote the modelled yield under rain-fed and irrigated conditions, respectively; $Area_{rain}$ and $Area_{irri}$ are the CFT-specific rain-fed and irrigated areas used in simulations, respectively (Fig. S3).

As climate input, monthly data at 0.5° resolution from 1901-2014 was taken from observation-based CRUJRA v2.1 (Harris et al., 2020; Kobayashi et al., 2015). Annual atmospheric $CO_2$ concentration over the same period was from the data set provided by Meinshausen et al. (2020). Land use/land cover information was used from LUH2 (Land-Use Harmonization 2; Hurtt et al., 2020) with fractions of natural vegetation, pasture, and cropland at each grid cell, spanning from 1901 to 2014 and remapped to the same resolution of climate forcing. The fractional cover of various crop species in the year 2000 came from MIRCA (Monthly Irrigated and Rain-fed Crop Areas; Portmann et al., 2010) and was aggregated to the six CFTs modelled in this study. Generally, the total cropland cover in a grid cell could change annually over time, but the relative fractions of each CFT within that cover fraction were held constant. In addition, the soil fractions of sand, silt, and clay in the topsoil (0-30 cm) from GGCMI phase 3 (Global Gridded Crop Model Intercomparison; Volkholz and Müller, 2020) were used to parameterize soil hydraulic properties at each modelled grid cell.



Monthly atmospheric N deposition ($NH_x$, $NO_y$) for 1901-2014 was used as simulated by CCMI (NCAR Chemistry-Climate Model Initiative). The value was interpolated to $0.5° \times 0.5°$ from the original resolution ($1.9° \times 2.5°$) to match the resolution of the climate data (Tian et al., 2018). In terms of N fertilizer input to the cropland, CFT-specific data for mineral N fertilizer and manure over 1901-2014 came from Ag-GRID (AgMIP GRIDded Crop Modeling Initiative; Elliott et al. (2015) and Zhang et al. (2017), respectively) (Fig. S3). Details on the fractions of mineral fertilizer applied to different crop development stages are provided in Table S1.

### 2.3.3 Ecosystem responses to management practices in Eastern Africa

To detect the effects of agricultural practices on food security and environmental sustainability regionally, five alternative management practices—N-fixing cover crop ($F_{CC-BNF}$), non-N-fixing cover crop ($F_{CC-NoBNF}$), residue retention ($F_{RR}$), manure application ($F_{Man}$), and no-tillage ($F_{NT}$)—together with an integrated management were assessed (Table 3); the latter integrated management with most individual practices included was selected to be representing conservation agriculture ($F_{conserv}$). Simulated outputs of these six management practices were compared with a 'standard' simulation ($F_{std}$) with set-ups shown in Table 3. The practice that produced the largest SOC increase at each grid cell was chosen as the optimal C management ($F_{opt}$) for the historical and future simulations:

$$F_{opt} = \{MAX\,(SOC_i - SOC_{F_{std}}), i = 1:5\} \tag{4}$$

where $F_{opt}$ is the calculated optimal (i.e., best performing) C management in a given grid cell; $i$ represents the five management practices of $F_{CC-BNF}$, $F_{CC-NoBNF}$, $F_{RR}$, $F_{Man}$, and $F_{NT}$. $SOC_i$ and $SOC_{Fstd}$ are the modelled SOC stocks from these five practices and standard management, respectively.

An initial experiment (B, Table 1) was performed to simulate the effects of these management practices under constant climate, $CO_2$, and land use in order to isolate management effects from environmental change impacts. This began with a run of the historical period (1901-2014) after model spin-up, using time-dependent gridded climate, land cover, and N inputs (deposition and fertilizer) at $0.5°$ resolution, combined with $CO_2$ concentration described in Sect. 2.3.2. The result of this run was to generate present-day cropland soil C and N pools under $F_{std}$ over Eastern Africa (B1, Table 1). Subsequent runs, one using each management practice, branched from this present-day state in 2015. In these simulations de-trended climate (repeating 1995-2014) and fixed $CO_2$ concentration (~397 ppm), together with N fertilizer and land cover data of the year 2014 were repeated for 86 years to allow soil C and N pools to reach a new equilibrium after the management shift (B2, Table 1). Our aim here was not to realistically reproduce the size of soil C and N pools in 2100 with different management practices, but rather to assess management long-term potential effects on crop ecosystems relative to the conventional practice (i.e., $F_{std}$). All simulated outputs in the last ten years of the model experiments were taken for analysis.

In a second experiment (C, Table 1), simulations were driven with future monthly climate data taken from five GCMs (Eyring et al., 2016), for 1901-2100 at $0.5°$ spatial resolution (see Table 1 for each GCM information). The climate data was used for the entire simulation period to avoid any inconsistency between the historic and future periods. For the historical period (1901-2014), the management set-up was the same as the simulation of B1 as described above, but with GCM-based climate forcings (C1, Table 1). The seven management practices listed in Table 3 started in the year 2015, with dynamic climate, $CO_2$ concentration, and N deposition throughout. Land cover and fertilizer use (mineral N and manure) were fixed from 2014 onwards to exclude their effects on cropland SOC sequestration (C2, Table 1). N deposition and climate data for SSP1-RCP2.6 (SSP1-26), SSP3-RCP7.0 (SSP3-70) and SSP5-RCP8.5 (SSP5-85) radiative forcing projections were selected due to the contrasting climate change and $CO_2$ concentration of the three scenarios (Meinshausen et al., 2020). Similar to the B2 simulation, the long-term management effects excluding environmental change impacts were also investigated using GCM-based repeated climate (C3, Table 1). The modelled SOC in the last ten years of the C2 simulation (2091-2100) was taken to compare with the C3 output over the same period in order to explore the

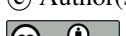



potential transition of the optimal C management ($F_{opt}$) caused by future climate change. Details on experimental set-ups are provided in Table 1.

260  **2.4 Data analysis**

The accuracy of the model in predicting SOC and yields was assessed using the coefficient of determination (adjusted $R^2$), relative bias (RB), absolute bias (AB), and the root mean square error (RMSE):

$$RB = \frac{M_i - O_i}{O_i} \times 100\% \tag{5}$$

$$AB = \frac{|M_i - O_i|}{O_i} \times 100\% \tag{6}$$

$$RMSE = \sqrt{\frac{1}{n}\sum_{i=1}^{n}(M_i - O_i)^2} \tag{7}$$

where $M_i$ and $O_i$ indicate modelled and observed values, $n$ is the number of observations. To evaluate the agreement of the interannual variability of modelled and reported yields in the long term, the Pearson correlation coefficient ($r$) were calculated:

$$r = \frac{\sum_{i=1}^{n}(M_i - \bar{M})(O_i - \bar{O})}{\sqrt{\sum_{i=1}^{n}(M_i - \bar{M})^2 \sum_{i=1}^{n}(O_i - \bar{O})^2}} \tag{8}$$

265  where $\bar{M}$ and $\bar{O}$ represent modelled and observed mean, and $n$ is the number of years.

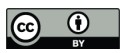

Table 1. Summary of simulations performed in this study. See methods section for abbreviations and further explanations.

| Purpose of simulation | Code | Trial involved | Time period | Model spin-up | Land-use | Climate | Manure input | Mineral N input | Number of simulations |
|---|---|---|---|---|---|---|---|---|---|
| SOC storage and maize yield evaluation against field-based trials | A1 | INM3 site | 1901-2015 | 500years Land use started 470 years PNV, then 30-years cropland ramp to 1901 $CO_2$ fixed in 1901 | 1901-2002: 100% grassland; 2003-2015: 100% cropland | GSWP3-W5E5 | 70 kg N ha⁻¹ | 0, 30, 60, and 90 kg N ha⁻¹ | 16 [a] |
| | A2 | CT1 site | 1901-2015 | The same as A1 | 1901-1991&1995-2000: 100% grassland; 1992-1994&2001-2015: 100% cropland | GSWP3-W5E5 | 0 kg N ha⁻¹ | 0, 30, 60, and 90 kg N ha⁻¹ | 16 [a] |
| Regional crop yields comparison & Response to different management practices, historical | B1 | $F_{std}$ (standard management) | 1901-2014 | The same as A1, baseline simulation for B2 | LUH2 | CRUJRA | Zhang et al. (2017) | Ag-GRID | 1 |
| | B2 | All managements [b] | 2015-2100 | Starting from B1 in 2014 | LUH2, fixed in 2014 | CRUJRA, 1995-2014 climate repeated until 2100 [c] | Zhang et al. (2017), fixed in 2014 | Ag-GRID, fixed in 2014 | 7 |
| Response to different management practices, future | C1 | $F_{std}$ (standard management) | 1901-2014 | The same as A1, baseline simulation for C2 and C3 | LUH2 | 5 GCMs [d] | Zhang et al. (2017) | Ag-GRID | 5 |
| | C2 | All managements [b] | 2015-2100 | Starting from C1 in 2014 | LUH2, fixed in 2014 | 5 GCMs × 3 SSP | Zhang et al. (2017), fixed in 2014 | Ag-GRID, fixed in 2014 | 105 |
| | C3 | All managements [b] | 2015-2100 | Starting from C1 in 2014 | LUH2, fixed in 2014 | 5 GCMs, 1995-2014 climate repeated until 2100 [e] | Zhang et al. (2017), fixed in 2014 | Ag-GRID, fixed in 2014 | 35 |

[a] For details of 16 simulations, see Table 2; [b] All managements denote the seven practices listed in Table 3; [c] Historical (CRUJRA-based) climate with temperature de-trended. These 20 years are repeated throughout the period 2015-2100; [d] Five GCMs——GFDL-ESM4, UKESM1-0-LL, MPI-ESM1-2-HR, IPSL-CM6A-LR, and MRI-ESM2-0——are used. GCM climate is bias corrected and statistically downscaled against observational dataset GSWP3-W5E5 (Lange, 2019) at daily temporal resolution at the entire globe; [e] The same as "c", but GCM-based historical climate.




270   **Table 2**. Site- and treatment-specific data used for model evaluation at the INM3 and CT1 long-term (2003-2015) trials. The "x" in the treatment names denotes any mineral N application rate of 0, 30, 60, and 90 kg N ha[-1]. The 33% and 66% of the mineral N-fertilizer are applied at the time of sowing and halfway of the vegetative stage, respectively, following the description in Sommer et al. (2018). Abbreviations: NoMan – no manure application; NoRR – no residue retention; NoTill – no-tillage; Man – 70 kg N ha[-1] of manure application converted from 4 t ha[-1] dry matter; RR – 50% of residue retention; Till – Tillage.

| Site and its soil physical properties | Treatment name | Tillage | Manure (kg N ha[-1]) | Residue retention (%) | Mineral N application (kg N ha[-1]) | | | | |
|---|---|---|---|---|---|---|---|---|---|
| | | | | | N app. timing | N0 | N30 | N60 | N90 |
| INM3 (34.40°E, 0.14°N) Topsoil (0-20cm): Sand: 26% Silt: 18% Clay: 56% Bulk density: 1.1 g cm[-3] | Nx_NoMan_NoRR | Yes | No | No | Sowing | 0 | 10 | 20 | 30 |
| | Nx_NoMan_RR | Yes | No | 50 | | | | | |
| | Nx_Man_NoRR | Yes | 70 | No | | | | | |
| | Nx_Man_RR | Yes | 70 | 50 | | | | | |
| CT1 (34.41°E, 0.13°N) Topsoil (0-40cm): Sand: 16% Silt: 15% Clay: 69% Bulk density: 1.1 g cm[-3] | Nx_NoTill_NoRR | No | No | No | Halfway of the vegetative stage | 0 | 20 | 40 | 60 |
| | Nx_NoTill_RR | No | No | 50 | | | | | |
| | Nx_Till_NoRR | Yes | No | No | | | | | |
| | Nx_Till_RR | Yes | No | 50 | | | | | |

275

**Table 3.** Simulation set-ups used for comparison of SOC sequestration, crop yields and cropland N losses with different managements over Eastern Africa for historical and future runs (see Sect. 2.3.3).

| Simulation [a] | $F_{CC-BNF}$ | $F_{CC-NoBNF}$ | $F_{RR}$ | $F_{Man}$ | $F_{NT}$ | $F_{conserv}$ | $F_{std}$ |
|---|---|---|---|---|---|---|---|
| N-fixing cover crop | Yes | No | No | No | No | Yes | No |
| Non-N-fixing cover crop | No | Yes | No | No | No | No | No |
| Residue retention | 10% | 10% | 100% | 10% | 10% | 100% | 10% |
| Manure application | Yes | Yes | Yes | No | Yes | Yes | Yes |
| Mineral N fertilizer | Yes | Yes | Yes | Yes | Yes | Yes | Yes |
| Tillage | Yes | Yes | Yes | Yes | No | No | Yes |

[a] Abbreviations: CC-BNF – N-fixing cover crop; CC-NoBNF – non-N-fixing cover crop; RR – residue retention; Man – manure application; NT – no-tillage; conserv – conservation agriculture; std – standard management.





## 3 Results

### 3.1 Model performance at site scale

The simulated maize yields in the long rainy season (from March to August) tended to be somewhat higher than the measurements, with the mean overestimation ranging from 18% to 21% at the two experimental sites (Fig. 1). The averaged yields over the entire experimental period (2004-2015) between simulations and observations compared well across all the evaluated treatments, with the simulated values falling within the range of measured standard deviation (Fig. S4). However, LPJ-GUESS did not capture the interannual variations of the yields well, producing a low Pearson correlation coefficient ($r$) and high absolute bias (AB) in all the INM3 and CT1 experiments (Table S2). As expected, measured and simulated yields in combined conservation managements (e.g., manure with residue retention at INM3) were higher than the individual ones in the little-fertilized treatments, but yield discrepancies between managements became small and insignificant when maize received a high N application rate of 90 kg N ha$^{-1}$ (Table S2).

The simulated SOC at both sites showed a declining trend from 2004-2015 under all the assessed treatments, agreeing well with the observation of soil C loss over the same period; however, the model generally underestimated SOC at the beginning of experiment while overestimating soil C stocks in the last two sampling years (Figs. 2 and 3). A linear correlation ($p<0.01$) between the simulated and measured SOC was found when all the managements were included, with the model explaining 82% and 64 % of the variation in observed SOC at INM3 and CT1, respectively (Fig. 1). Low absolute bias of 4.2% and RMSE value of 4.1 t C ha$^{-1}$ were found for the INM3 treatments, and 3.5% and 3.9 t C ha$^{-1}$ for the CT1 treatments (Fig. 1). The field-measurements showed that SOC stocks from the combined conservation managements were significantly higher than the conventional ones (i.e., Nx_NoMan_NoRR at INM3 and Nx_Till_NoRR at CT1). The model can generally capture this response well, but it had difficulty in predicting SOC difference between the individual managements at both sites (Figs. 1-3).



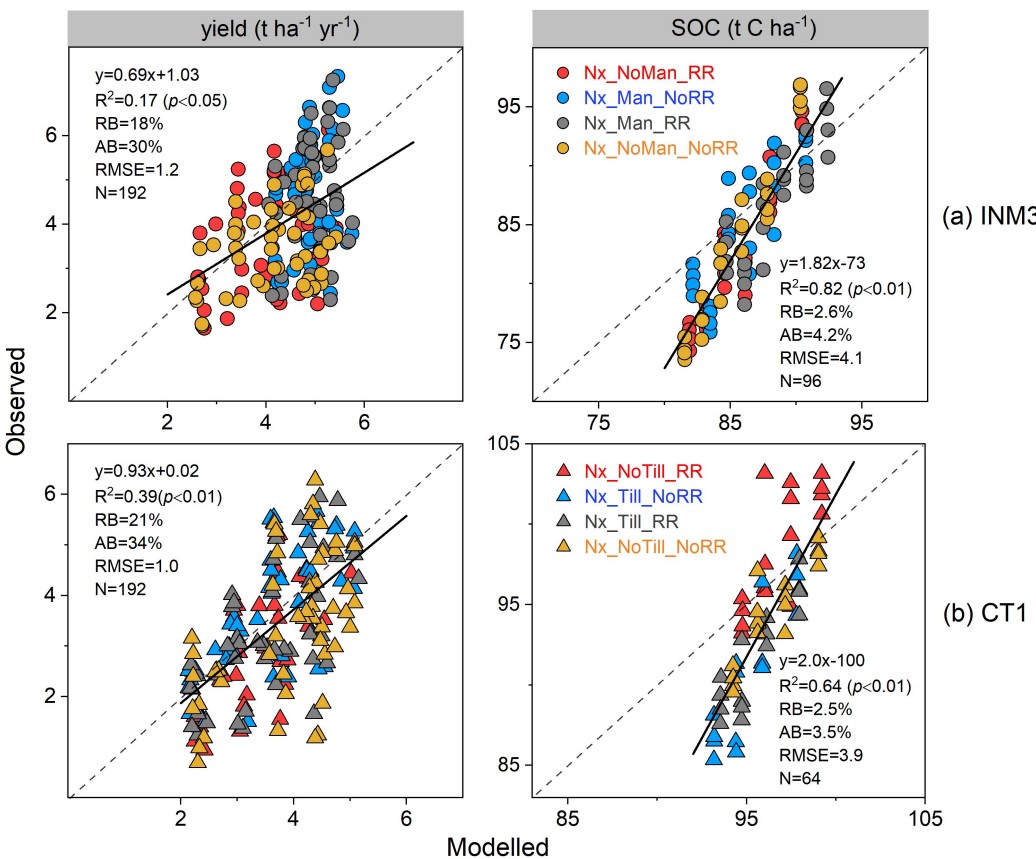

**Figure 1.** Comparison of modelled and observed maize yields (long rainy season, i.e. the main growing period) and SOC stocks at INM3 (a) and CT1 (b) sites for all treatments listed in Table 2. The dashed line is 1:1 line and black bold line is fitted linear regression; RB and AB are relative bias (Eq. 5) and absolute bias, (Eq. 6), respectively, representing in percent (%); RMSE is root mean square error, with the unit of t ha$^{-1}$ yr$^{-1}$ for yield and t C ha$^{-1}$ for SOC. See Table 2 for the treatment abbreviations and their explanations.



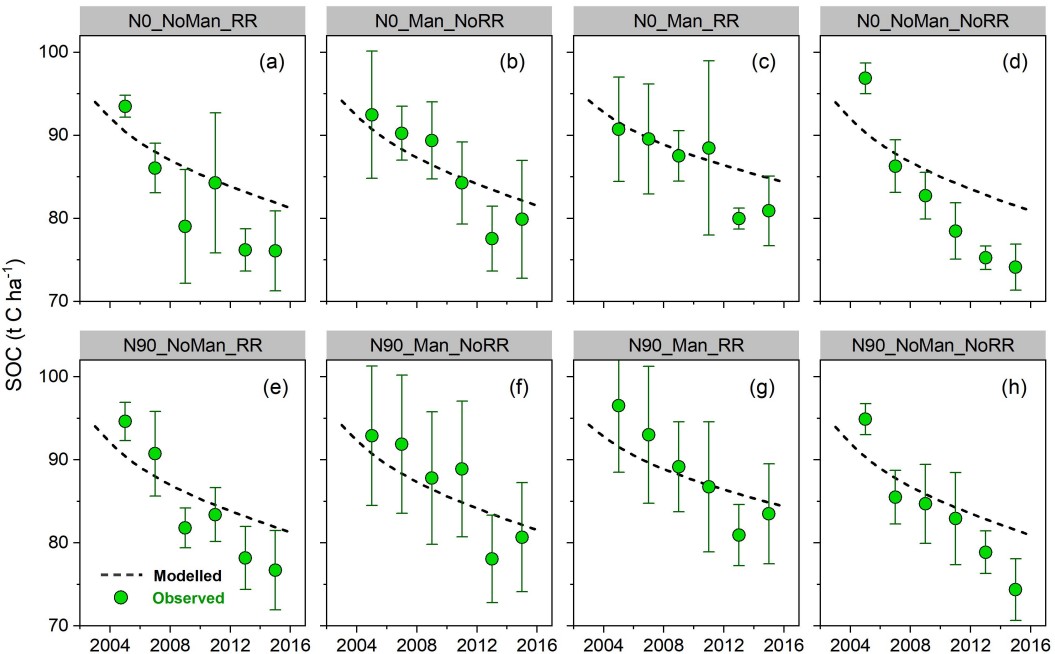

**Figure 2.** The modelled and observed SOC stocks (0-150cm) for the evaluation treatments (a-h) at the INM3 site, with two levels of mineral N fertilizer input (N0 and N90). The dashed line is the modelled SOC, and the closed circle represents the observed value (scaled to 150 cm depth) averaged by the four replicates in the trials, with standard deviation as given in the vertical bar.

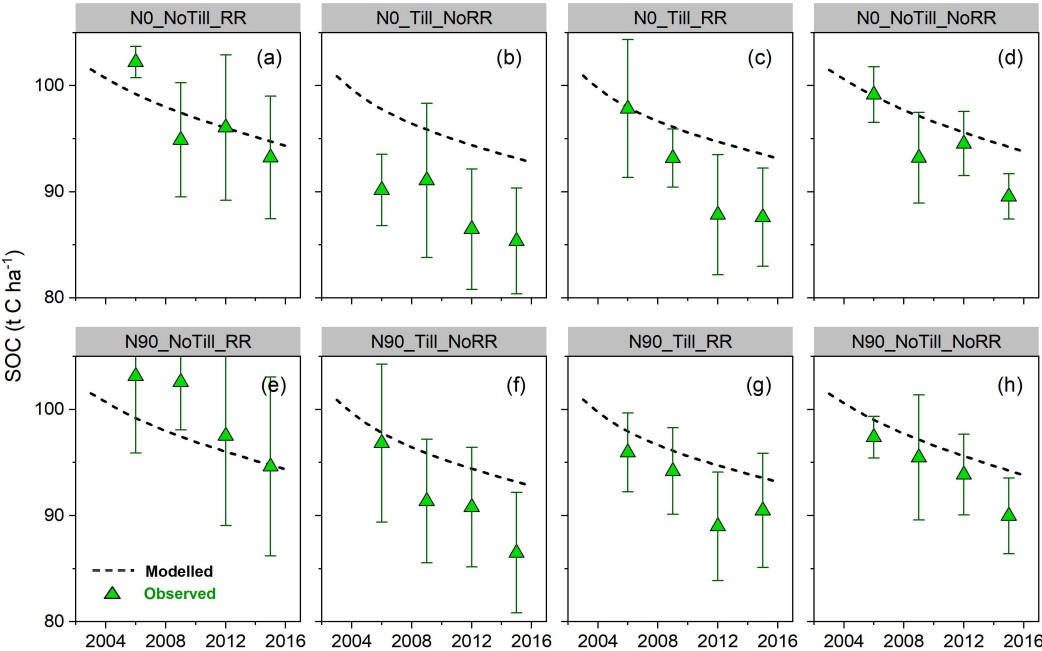

**Figure 3.** The same as Fig.2, but at the CT1 site. See Table 2 for the treatment abbreviations and their explanations.



Compared to observations, LPJ-GUESS underestimated absolute SOC loss in the INM3 experiments (Table 4). Due to the extra C input to soils from manure and residue retention, the simulated combination of these two managements yielded the lowest loss of 6.7 t C ha$^{-1}$ at INM3 site (N0_Man_RR), with this loss reduced by mineral N addition (6.3 t C ha$^{-1}$, N90_Man_RR). By contrast, the model produced the largest C loss of 8.9 t C ha$^{-1}$ in the unfertilized maize control treatment with no manure and residue application (N0_NoMan_NoRR). This estimate was also similar to the simulated loss of 8.6 t C ha$^{-1}$ in maize residue only (N0_NoMan_RR) and

manure application only (N0_Man_NoRR) practices. In general, the application of manure and residue retention, together with 90 kg N ha$^{-1}$ of fertilizer input was modelled to reduce SOC loss by 29% in comparison with the control treatment, lower than the observed reduction of 43% (Table 4).

    Similar to INM3, the observed absolute SOC loss across all the CT1 treatments was underestimated but with smaller absolute differences (Table 4). As expected, LPJ-GUESS simulated a high C loss of 4.6 and 4.5 t C ha$^{-1}$ in the tilled cropping systems with no

maize residue retained (N0_Till_NoRR and N90_Till_NoRR). Implementing minimum tillage reduced this loss to 4.4 and 4.3 t C ha$^{-1}$ in treatments with residue retention included (N0_NoTill_RR and N90_NoTill_RR). Compared to the control treatment (N0_Till_NoRR), the simulated application of maize residue with 90 kg ha$^{-1}$ of N-fertilizer reduced SOC loss by 2% (N90_Till_RR), while adopting minimum tillage could further reduce the loss by 7% (Table 4).

**Table 4.** Comparison of modelled and observed SOC stocks and absolute SOC loss for all the evaluated treatments over 2005-2015 at the INM3 and

CT1 sites. The absolute SOC loss of each treatment was calculated as the difference between the first (year 2005 and 2006 for the INM3 and CT1 trials, respectively) and last sampling year (i.e., 2015). The observed SOC are represented as mean ± 1 standard deviation, deriving from the four replicates in each treatment.

| | SOC (t C ha$^{-1}$) | | | | Absolute SOC loss (t C ha$^{-1}$) | |
|---|---|---|---|---|---|---|
| | Year 2005 (or 2006) | | Year 2015 | | 2015 minus 2005 (or 2006) | |
| | Observed | Modelled | Observed | Modelled | Observed | Modelled |
| **INM3 site** | | | | | | |
| N0_NoMan_RR | 93.5±1.3 | 90.5 | 76.1±4.8 | 81.9 | -17.4 | -8.6 |
| N0_Man_NoRR | 92.5±6.7 | 90.8 | 79.9±7.1 | 82.2 | -12.6 | -8.6 |
| N0_NoMan_NoRR | 96.9±1.9 | 90.3 | 74.1±2.8 | 81.4 | -22.8 | -8.9 |
| N0_Man_RR | 90.7±6.3 | 91.6 | 80.9±4.2 | 84.9 | -9.8 | -6.7 |
| N90_NoMan_RR | 94.6±2.3 | 90.5 | 76.7±4.8 | 82.0 | -17.9 | -8.5 |
| N90_Man_NoRR | 92.9±8.4 | 90.9 | 80.7±6.6 | 82.4 | -12.2 | -8.5 |
| N90_NoMan_NoRR | 94.9±1.9 | 90.4 | 74.4±3.7 | 81.6 | -20.5 | -8.8 |
| N90_Man_RR | 96.5±8.0 | 91.5 | 83.5±6.0 | 85.2 | -13.0 | -6.3 |
| **CT1 site** | | | | | | |
| N0_NoTill_RR | 102.2±1.5 | 99.2 | 93.2±5.8 | 94.8 | -9.0 | -4.4 |
| N0_Till_NoRR | 90.2±3.4 | 97.8 | 85.3±4.9 | 93.2 | -4.9 | -4.6 |
| N0_Till_RR | 97.8±6.5 | 98.0 | 87.6±4.6 | 93.5 | -10.2 | -4.5 |
| N0_NoTill_NoRR | 99.2±2.6 | 99.0 | 89.6±2.1 | 94.2 | -9.6 | -4.8 |
| N90_NoTill_RR | 103.2±7.3 | 99.4 | 94.6±8.4 | 95.1 | -8.6 | -4.3 |
| N90_Till_NoRR | 96.8±7.5 | 97.8 | 86.5±5.7 | 93.3 | -10.3 | -4.5 |
| N90_Till_RR | 95.9±3.7 | 98.1 | 90.5±5.4 | 93.6 | -5.4 | -4.5 |
| N90_NoTill_NoRR | 97.4±1.9 | 99.1 | 89.9±3.6 | 94.5 | -7.5 | -4.6 |



### 3.2 Regional yields comparison

Using the CFTs-specific parameters given in Table S1, combined with the time-dependent gridded N-fertilizer data set introduced in
Sect. 2.3.2, we simulated crop yields in Eastern Africa under the standard management ($F_{std}$, Table 3) from 1901-2014. The modelled
outputs from 1961-2014 and 1993-2014 were chosen to compare with annual FAO yields in Kenya and Ethiopia, respectively, due to
their different time frames reported in statistics.

Modelled maize yields in the two countries showed a good agreement with observations, with a low relative bias (RB) of -6 % and
RMSE value of 0.22 t ha$^{-1}$ yr$^{-1}$ in Kenya, and -21% and 0.54 t ha$^{-1}$ yr$^{-1}$ in Ethiopia (Figs. 4a-b). LPJ-GUESS tended to overestimate the
reported yields in pulses and sorghum, with the country-level overestimation spanning from 48-257% and 72-203%, respectively.
With the exception of sorghum in Kenya, the correlation between the simulated and reported yields were positively significant in most
crop types, with a Pearson correlation coefficient ($r$) from 0.55-0.90 (p<0.001, Figs. 4a-b), indicating that the model was able to
capture the interannual variability in yields despite some deviations from observations for individual years. Also, the total modelled
maize production on regional scale increased from 5.4 million tonnes (Mt) in 1993 to 9.8 Mt in 2014, in line with the reported values
of 3.6-11.2 Mt yr$^{-1}$ over the same period (Fig. 4c). Including all six agricultural crops in LPJ-GUESS gave a mean total production of
19.7 Mt yr$^{-1}$from 1993-2014, ~45% higher than the FAO statistics of 13.6 Mt yr$^{-1}$, mainly due to the large overestimation in pulses
production regionally (6.6 and 2.2 Mt yr$^{-1}$ for modelled and reported yields, respectively).

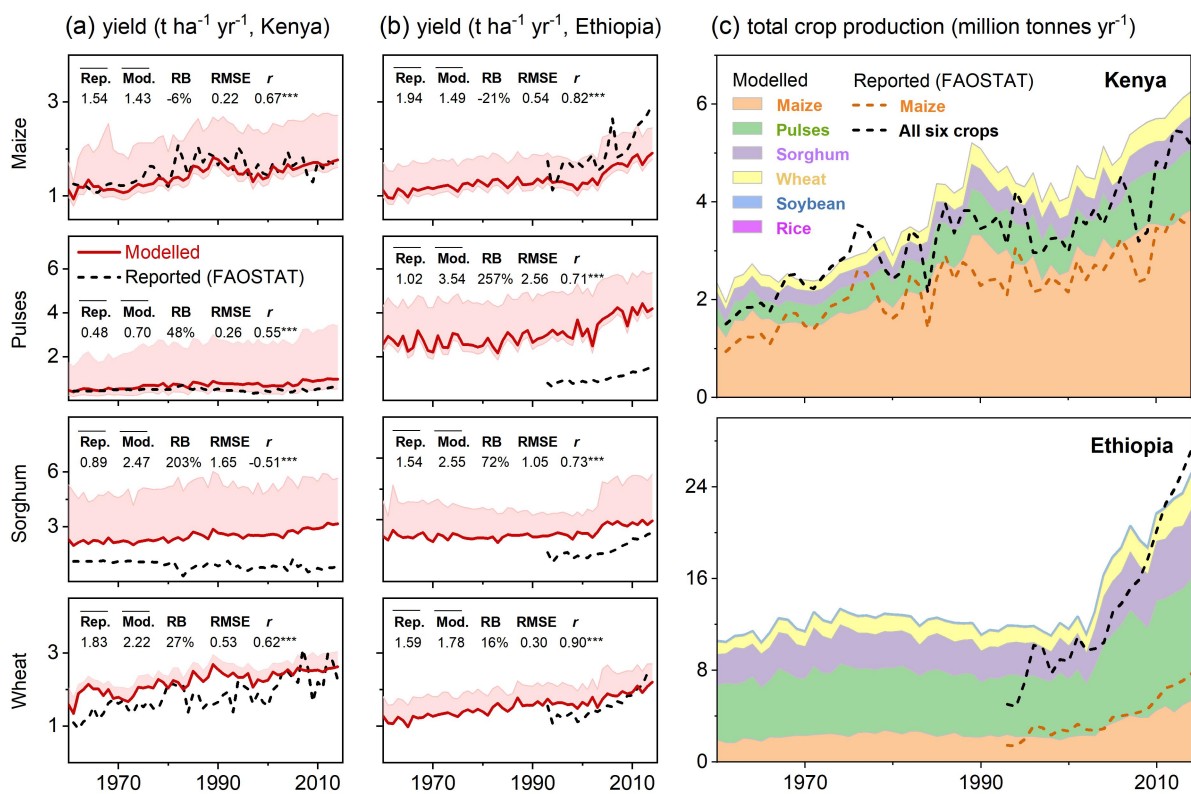

**Figure 4.** Comparison of modelled and FAO-reported crop annual yields on country level from 1961-2014 in Kenya (a), Ethiopia (b), and total crop
production (c). The upper and lower bounds of shade areas in (a) and (b) represent the simulated yields in irrigated and rain-fed conditions,
respectively, with their area-weighted aggregated results as given in red solid lines. $\overline{Rep.}$ and $\overline{Mod.}$ indicate the reported and modelled yields
averaged over FAO-based periods (1961-2014 for Kenya and 1993-2014 for Ethiopia), respectively; RB is relative bias, represented in percent (%);




RMSE is root mean square error, with the same unit as yield (t ha⁻¹ yr⁻¹); *r* is Pearson correlation coefficient, where *** denote the correlation to be statistically significant at *p*=0.001 level.

### 3.3 Ecosystem responses to management practices in Eastern Africa

#### 3.3.1 Historical runs

With six crop types included, all the explored management options that address aspects of sustainable land management resulted in a net increase in simulated cropland SOC in Kenya and Ethiopia compared to the standard management (Fig. 5a). As expected, our simulation of the integrated conservation agriculture practice generated nearly the largest increase in soil C sequestration of ~11%, followed by cover crops (both N-fixing and non-N-fixing), residue retention, and manure application, with the lowest increase of ~2% found in no-tillage management practice. Most of these investigated practices also achieved the extra benefit of increased yields—despite being in our simulations accompanied by larger N losses—with the exception of cover crops in some regions. Compared to a non-N-fixing cover crop, the implementation of a N-fixing cover crop was modelled to broadly produce higher N loss over Eastern Africa. However, this practice was accompanied by an increase in simulated yields of ~18%, as a result of additional N input through symbiotic N fixation in herbaceous legumes, which facilitates a N-rich soil environment to subsequent crops for better growth and productivity. Leaving most parts of crop residues in the field and applying manure as fertilizer were the two treatments that increased the modelled yields for most croplands but with the large environmental "cost" of an increase in N loss. The increase in both yield and N loss from residue retention likely reflects that N becomes available for crop uptake over a longer period, and nothing grows between the growing periods which can increase the N leaching from soil. In addition, no-tillage, as an important component in conservation agriculture in the tropics of Africa, was simulated to potentially reduce the N loss from cropland with slight yield benefits depending on region (Fig. 5a) and cropping system (Fig. S5).

The impacts of individual (and combined) management techniques varied widely between different parts of Kenya and Ethiopia, depending on climate and soil condition, as well as crop types (Fig. S5). In general, N-fixing cover crop was identified as a promising option for potentially sequestering SOC, with 43% of cropland grid cells having this practice as the optimal C management ($F_{opt}$), followed by manure, residue retention, and the standard management practice ($F_{std}$, Fig. 5b). However, this spatial pattern showed distinct difference between crop types. For instance, incorporating crop residue to soil was simulated to dominate soil C responses in maize and sorghum systems, but it only slightly contributed to SOC enhancement in wheat and pulses cropping systems (Fig. S6), likely reflecting their differences in biomass production, phenological responses to climate change and N-fertilizer application rates (Fig. S3).

The simulated cropland soil C stock (0-150 cm) from various managements ranged from 932-1038 Tg C in Kenya and 2569-2895 Tg C in Ethiopia, which, as expected, was larger than the published sources for the depth layer 0-30 cm (Zomer et al., 2017). However, these modelled soil C stocks compared reasonably with the scaled-up published estimates based on the depth distribution functions (Eqs. 1 and 2), with 727 and 2227 Tg carbon estimated for the depth of 0-150 cm in Kenya and Ethiopia, respectively (Table 5). The total simulated maize production of 8.7-14.3 Mt yr⁻¹ on regional scale was close to the FAO-reported yield of 11.2 Mt yr⁻¹. With all agricultural crops included, an overall overestimation of 7-47% was found (Table 5), primarily reflecting the overestimated production in pulses and sorghum described in Sect. 3.2 (Fig. 4).

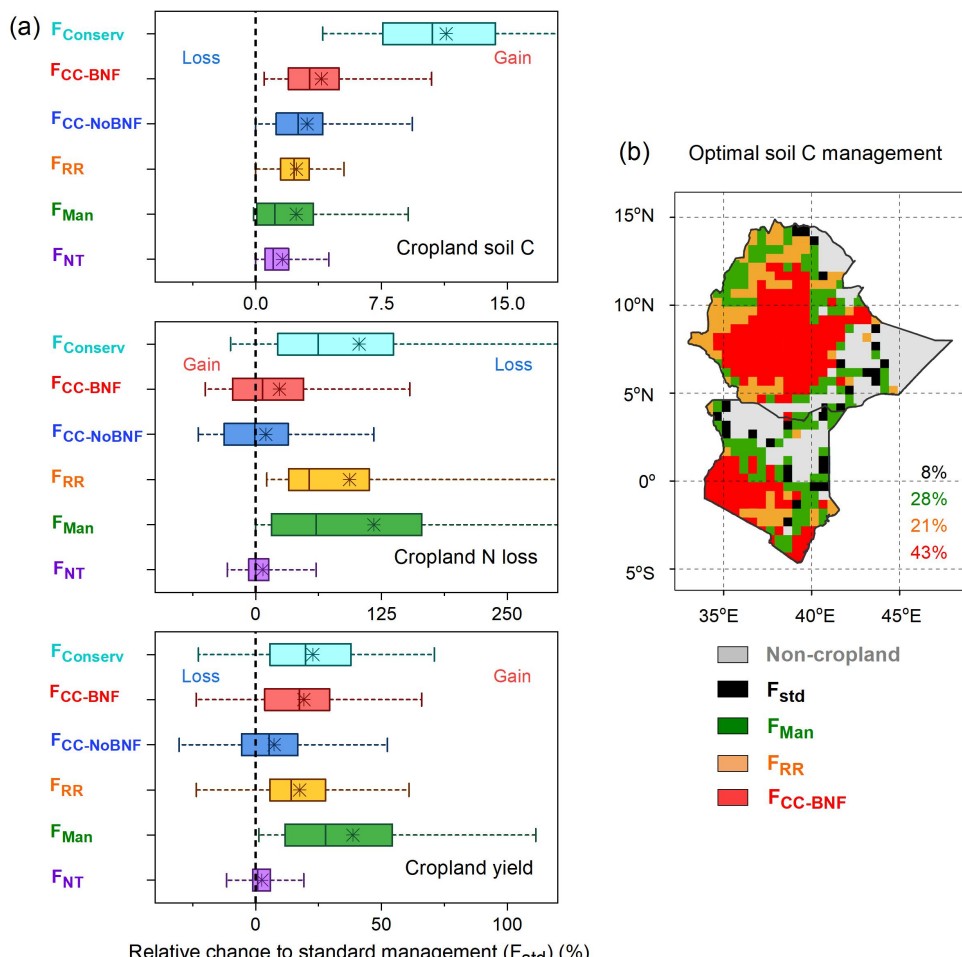


**Figure 5.** The modelled relative response (%) of cropland SOC, N loss, and yield to alternative management techniques (see Table 3 for abbreviations) compared to the standard management prevalent in the area (a), and the optimal C sequestration practice ($F_{opt}$, Eq.4) simulated by LPJ-GUESS in Kenya and Ethiopia (b). Box plots in (a) denote the 5th and 95th percentiles with whiskers, median and interquartile range with box lines, and mean with asterisks across all cropland grid cells (428). The numbers in (b) represent the grid cell proportion of each optimal management from 390 the total grid area. The standard management ($F_{std}$, black in (b)) was chosen when none of other alternative managements yielded a net increase in SOC.





**Table 5.** Modelled total cropland soil C stocks (0-150 cm), N loss, and total crop production with different management options in Kenya and Ethiopia, compared to literature-based estimates. See Table 3 for abbreviations.

| Management | Soil C stock, total (Tg C) | | N loss (Gg N yr⁻¹) | | Crop production (million tonnes yr⁻¹) | | | |
|---|---|---|---|---|---|---|---|---|
| | Kenya | Ethiopia | Kenya | Ethiopia | Kenya | | Ethiopia | |
| | | | | | Maize | All crops [a] | Maize | All crops [a] |
| $F_{std}$ | 939 | 2592 | 61 | 157 | 3.9 | 7.3 | 7.3 | 21.7 |
| $F_{NT}$ | 948 | 2623 | 64 | 164 | 3.8 | 7.6 | 7.1 | 23.4 |
| $F_{Man}$ | 932 | 2569 | 45 | 79 | 3.2 | 6.7 | 5.5 | 19.9 |
| $F_{RR}$ | 957 | 2653 | 134 | 359 | 4.2 | 8.4 | 8.2 | 26.7 |
| $F_{CC-NoBNF}$ | 969 | 2696 | 64 | 190 | 3.9 | 7.7 | 7.7 | 23.1 |
| $F_{CC-BNF}$ | 979 | 2710 | 75 | 204 | 4.9 | 8.9 | 8.5 | 24.7 |
| $F_{opt}$ | 993 | 2786 | 81 | 229 | 4.6 | 9.0 | 8.3 | 25.8 |
| $F_{conserv}$ | 1038 | 2895 | 127 | 375 | 5.1 | 9.4 | 9.2 | 27.0 |
| Other studies | 414[b] 727[c] | 1268[b] 2227[c] | 111 (76-297)[d] | — | 3.5[e] | 5.2[e] | 7.7[e] | 19.6[e] |

[a] Summed yield of six crop types: maize, pulses, sorghum, wheat, rice, and soybean; [b] Zomer et al. (2017); [c] Zomer et al. (2017), soil C stocks were scaled up to 0-150 cm from the original depth of 0-30 cm using the depth distribution functions (see Eqs. 1 and 2); [d] Zhang et al. (2021), the mean estimate over 2006-2015 was chosen, with a range given in bracket; [e] FAOSTAT (2021), the reported total production in the year 2014 were used for comparison, since the simulated cropland area was fixed from 2014 onwards (6,222,100 and 17,433,400 ha for Kenya and Ethiopia, respectively), see B2 in Table 1.

**3.3.2 Future projection**

Compared to the standard model set-up ($F_{std}$), all management practices were simulated to enhance the C storage in agricultural soils at the end of this century (i.e., 2091-2100) but with insignificant differences between three future climate change and $CO_2$ scenarios (Fig. 6a). Although no-tillage had nearly no impact on crop production, it was accompanied by the environmental benefit of N loss reduction (Fig. 6b). A clear yield difference between the three SSP scenarios was consistently seen in experiments with N-fixing cover

crop and conservation agriculture practices, with production increases being higher for SSP5-85 than for SSP1-26 climate pathway (Fig. 6c). This likely reflects the stronger $CO_2$ fertilization effect on the growth of herbaceous legumes under SSP5-85. Overall, the future projection showed that N-fixing cover crop represented a near win-win situation in the sense of SOC enhancement and yield increase in Eastern Africa, also with lower N loss compared to manure and residue application practices.

The simulated cropland soil C stocks (0-150 cm) under future conditions varied widely between the assessed management options,

with the integrated conservation agriculture being the only practice that showed positive soil C sequestration over the simulation period (2015-2100). Adoption of N-fixing cover crops contributed to increasing SOC stocks in the first two decades of the model experiments, after which stable SOC for SSP1-26 and slight C loss for SSP3-70 and 5-85 scenarios were simulated (Fig. S7). Other practices, such as the standard management and no-tillage, generally exhibited an obvious declining trend in total C storage between 2015 and 2100. Also, there were substantial changes in the optimal C sequestration practice for the future scenarios, with ~30% of

cropland areas in Kenya and Ethiopia (Table S3) showing the potential transitions in the last ten years of this century in comparison to



the present-day climate (GCM-based historical simulation; see C3, Table 1). Most of these shifts were simulated to come from the other management type options to N-fixing cover crop, such as manure application and residue retention (Fig. 7).

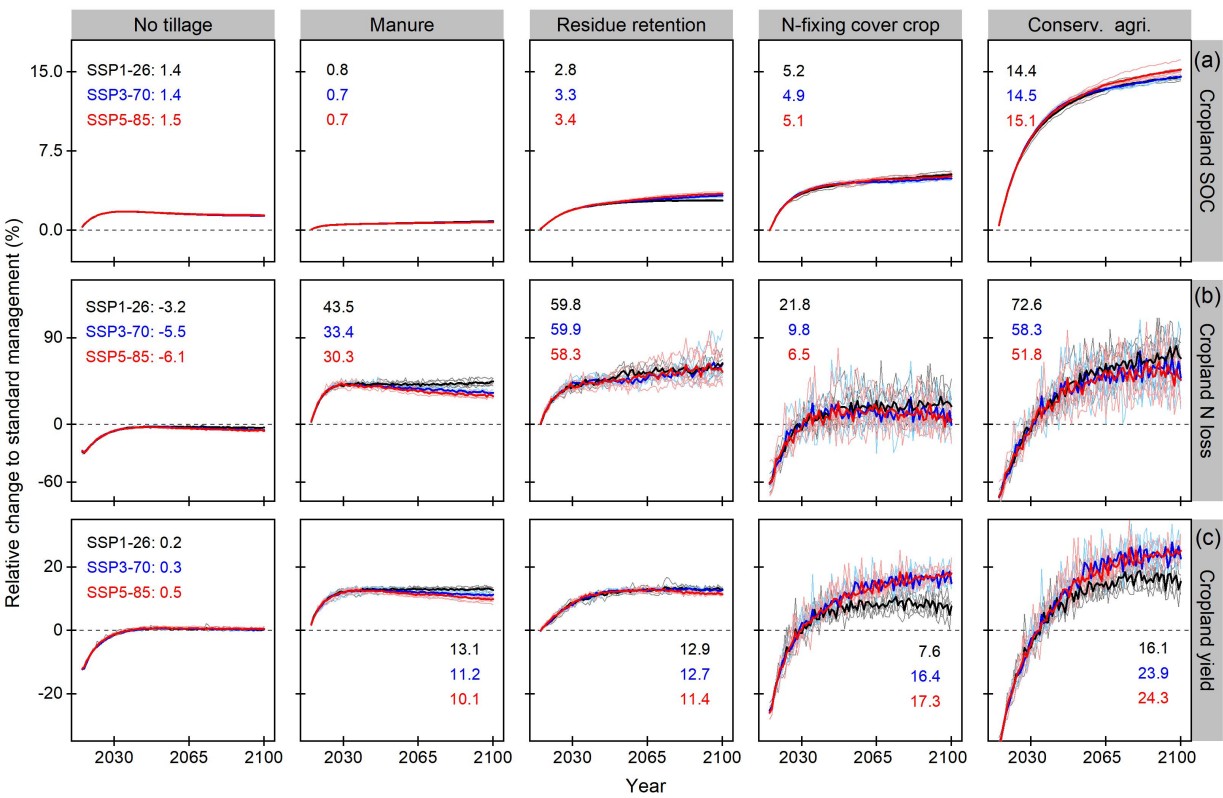

**Figure 6.** The simulated response (%) of cropland SOC (a), N loss (b) and yield (c) to alternative management techniques, relative to the model
standard management ($F_{std}$). The dark black, blue and red lines denote the mean of simulations using five GCMs (see Table 1) for SSP1-26, 3-70 and 5-85 scenarios, respectively. Lines in light color represent the simulation driven by individual GCMs. Numbers in plots indicate the averaged results between 2091 and 2100.

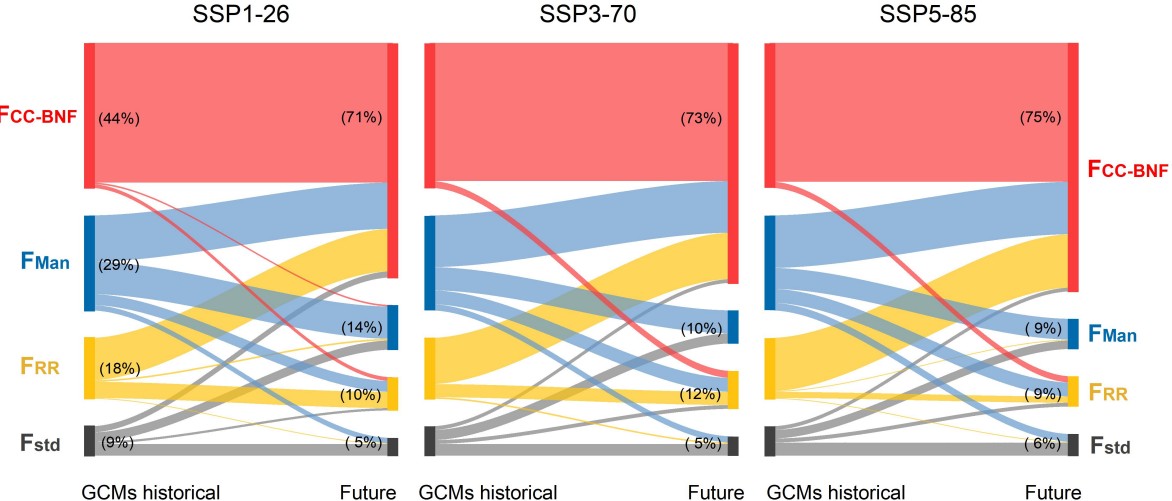

**Figure 7.** The relative (%) number of cropland grid cell regarding their optimal C sequestration practice (Eq. 4) for the historical period (C3, Table 1) and three future SSP scenarios from 2091-2100. The numbers in the figure represent the mean of simulations using five GCMs, and transitions indicated by the coloured bands. See Table 3 for management abbreviations.

## 4 Discussion

### 4.1 Uncertainties on model evaluation at site scale

LPJ-GUESS simulates the average maize yields among treatments over the experimental period well (2004-2015; see A1 and A2 in Table 1), but the measured interannual variability of the yields for the different management treatments was not well predicted. This issue is not unique to our study, and has also been found at the same sites using the DayCent model (Nyawira et al., 2021). The poor performance in modelling yield variability is likely due to the precipitation discrepancy between the gridded climate input data (i.e., GSWP3-W5E5) and field-based weather records (Fig. S1), resulting in the effects of extreme weather events (e.g., drought, rainstorms or flooding) being difficult to account for. Also, these impacts of extremes on physiological processes such as flowering or grain filling so far are not well represented in crop models, including LPJ-GUESS, but known to cause yield losses (Olin et al., 2015a; Nyawira et al., 2021).

Since LPJ-GUESS at this point does not simulate multi-cropping within a year, absence of maize residue and manure application events in the second cropping season (i.e., the short rainy season from September until January in western Kenya) may contribute to the underestimation of the measured SOC in the treatments with these two practices included (Table 4). In addition, compared to the fixed amount of maize residue retained in the field trials (2 t ha$^{-1}$), using 50% of residue retention in the model set-up is not equivalent to these stable and continuous C inputs to soils because of the varying biomass of simulated maize residue between years. This may partially explain the differences in the rates of SOC loss between the observed and simulated values at both sites.

All investigated management practices led to decline in SOC stocks in the field trials (Figs. 2 and 3), the overall trends were also reproduced by the model. Nonetheless, the negative soil C sequestration rates from 2004-2015 were unexpected, since addition of farmyard manure and residues can enhance SOC storage via additional C inputs to soils while conservation tillage slows down decomposition in the SOM pools. Both INM3 and CT1 sites in this study were under natural grassland before the trials start (see A1 and A2, Table 1), hence SOC losses in observations and simulations reflected a) grassland soils tending to store more carbon than cropland, and b) a new SOC equilibrium may not have been reached in the maize cropping systems after 10+ years of cultivation (Lal,





2008). A similar finding was reported by Moebius-Clune et al. (2011), who showed declining SOC in western Kenya even after more than 50 years of conversion from primary forest to maize. Furthermore, fast turnover of the SOM in the humid tropics could be another factor affecting the SOC trends because of the prevailing warm and moist climate (Olin et al., 2015a). The turnover-driven C losses at the sites may exceed the gains resulting from the C addition from manure and residue application (Kihara et al., 2020; Nyawira et al., 2021).

Agreement between the observed and simulated SOC declines was reasonable for all the considered treatments, although LPJ-GUESS generally underestimated the rates of SOC loss at the two experimental sites (Table 4). Previous studies have shown that high termite activity in western Kenya can strongly promote litter decomposition rates in the non-tilled maize cropping system (Ayuke et al., 2011; Kihara et al., 2015). We do not know whether this particular process played an important role at the field trials, but it is not included in the representation of SOM decay in the model (see Sect. 2.1). In principle, decomposition by soil animals could be addressed by

adjusting the decomposition parameters in the structural and metabolic litter pools (Nyawira et al., 2021), but adopting such an approach is currently prevented by the lack of information for evaluation.

    In order to compare the modelled SOC stocks with observations, as described in Sect. 2.3.1, we scaled up the measured SOC in the upper soil (0-15 cm) to the modelled depth of 150 cm using a simple extrapolation function. However, the extrapolated SOC values are most likely different from observations for the depth of 0-150cm because of the varying management effects on SOC changes with

depth. For instance, a recent analysis indicated that an intermediate and high intensity of tillage can significantly reduce SOC storage in agricultural soils, but large variations existed between soil layers (Haddaway et al., 2017). Scaling the SOC values with depth in the analysis cannot reflect this variability, and introduces uncertainties on soil carbon estimates in our evaluation.

### 4.2 Regional yields comparison

    Our simulated maize production at country level agreed well with FAO statistics in Kenya and Ethiopia, but a general yield

overestimation was found for most other crop types (Fig. 4). One factor contributing to the overestimation is that LPJ-GUESS applies a harvest efficiency of 90% to adjust the modelled crop yields on large spatial scales (Lindeskog et al., 2013). This value has been chosen to account for the crop post-harvest losses arising from mechanical and/or manual damage during harvest operation, or poor handling and/or storage conditions (Sheahan and Barrett, 2017; Stathers et al., 2020). The FAO (2011) reports that the quantity loss for cereals varies widely between regions due to the differences in management technology, ranging from 5-7% in Europe and North

America to 18% in sub-Saharan Africa (SSA). If the reported losses for SSA apply also to Kenya and Ethiopia, the value of 90% implemented in the model would lead to a yield overestimation by 10% regionally.

    A strong overestimation in pulses production was seen for both countries (Fig. 4). This can be mainly explained by the high legume N fixation capacity modelled by LPJ-GUESS in warm and moist climate (Ma et al., 2021). A high rate of BNF may reduce the N constraints on leaf photosynthesis and subsequently strengthen the flow of carbon assimilation to storage organ, resulting in the high

production in N-fixing crops. Yet, similar to pulses, our simulated sorghum yields at country level were also significantly greater than FAO records (Fig. 4). This suggest that other factors are at play as well. For example, insect pests, particularly shoot flies and stalk borers, have been identified as the major constraint to sorghum production in SSA (Wortmann et al., 2009), with estimated yield reduction of 11-49% in western Africa and 15-88% in Eastern Africa (Okosun et al., 2021). The present crop module of LPJ-GUESS does not take pests into account, which could contribute to the large overestimation of sorghum production in our studied region.

Additionally, a good representation of photosynthate allocation to various plant organs is especially important when modelling crop yields (Bondeau et al., 2007). In this study we updated the daily assimilate partitioning scheme of sorghum based on the existing literature (Fig. S2), but this process has not yet been parameterized and calibrated against observations from field experiments. Whether or not this related to the large-scale yield overestimation needs to be further investigated in future work.



### 4.3 Ecosystem responses to management practices in Eastern Africa

**4.3.1 Soil carbon stocks**

Published estimates of management improvement effects on the potential SOC increase on Kenya and Ethiopia cropland vary between 15.5-32.7 Tg yr$^{-1}$ assuming that the improved managements are continuously practiced over 20 years (Zomer et al., 2017). Across the eastern African study region, LPJ-GUESS predicted a SOC increase of 2.9 Tg yr$^{-1}$ for the optimal C management ($F_{opt}$) and 4.7 Tg yr$^{-1}$ for the integrated conservation agriculture practice ($F_{conserv}$) compared to the standard management ($F_{std}$, Table 5). The difference

between the estimates in Zomer et al. (2017) and our study may well be caused by our longer simulation period. When a change in management causes soil C stock to increase, it moves towards a new equilibrium value over a period of years or decades depending on climate and soil type (Johnston et al., 2009; Sommer and Bossio, 2014). In the early years after the change in management the annual rate of increase is largest, and it then gradually declines when the new SOC equilibrium value is approached (Poeplau and Don, 2015; Powlson et al., 2016). The 86 years of simulations in our study is roughly four times longer than the 20 years studied in Zomer et al.

(2017), and hence lower simulated annual rates of SOC increases are expected. If we consider the rates of SOC change over the first 20 years of simulation, the modelled soil C increase of 13.6 Tg yr$^{-1}$ from $F_{conserv}$ practice (not shown) is close to the lower end of the range found in Zomer et al. (2017).

Most CA techniques adopted in sub-Saharan Africa (SSA) are the combined treatments of minimum tillage and residue retention (Thierfelder et al., 2013; Cheesman et al., 2016). At the regional level, our modelled small SOC increase of 2% in no-tillage ($F_{NT}$) and

3% in residue retention ($F_{RR}$) agree with a recent meta-analysis of Githongo et al. (2021), in which converting from a conventional tillage to a no-till system in SSA on average showed only slight SOC increase in a maize cropping system. This reported insignificant impact contrasts with an earlier synthesis conducted by Powlson et al. (2016), who reported that the combination of minimum tillage and residue retention in SSA would result in a net SOC increase of 0.45 t C ha$^{-1}$ yr$^{-1}$ after three to nine years of implementation, ~24% higher than the control management (i.e., tillage + residue removal). In our model experiments, only the integrated conservation

agriculture practice ($F_{conserv}$) results in a fairly large SOC increase of 11% (varying from 4-22%, Fig. 5a), more comparable but still below the findings in Powlson et al. (2016). The reason for the disagreement between the regional simulation and field-based experiments is difficult to assess because of the difference in the studied geographical scales, land use history, sampled soil depth and implemented duration of practices. Nevertheless, they point robustly to the potential of affecting soil C storage positively through management, even though the magnitude remains unresolved.

Reflecting poor soil condition and limited manure availability, cropland SOC stocks are generally proportional to the applied amount of manure in SSA (Gross and Glaser, 2021). In our study, the regional mean application rate of manure from the year 2014 is modelled to vary from 12-32 kg N ha$^{-1}$ yr$^{-1}$ among crop types (Fig. S3b), resulting in an overall SOC increase of 3%, with a range of 0.2-9.1% depending on grid cell ($F_{Man}$, Fig. 5a). This simulated increase is comparable with results from the field experiments. For instance, in a four-year trial, Alemu and Bayu (2005) found that the SOC in Ethiopian sorghum fields with 21 and 42 kg N ha$^{-1}$ yr$^{-1}$ of

manure inputs were 7.8% and 9.4% higher than the control treatment, respectively. A similar SOC increase of 8-11% was also reported under maize systems with residue removal in western Kenya, but with 140 kg N ha$^{-1}$ yr$^{-1}$ organic fertilizer applied for 12 years (Sommer et al., 2018). In our study we implemented uniform C:N ratio of manure for all the simulated years and grid cells based on literature values (see Sect. 2.2.3). These set values cannot reflect the known considerable variation in C:N between manure types and locations in Eastern Africa that arise from different plant species consumed by the farm animals (Paul et al., 2009) and from

different farm animal species (Zhu et al., 2020). Absence of spatial variations in C:N ratio could bias the amount of C added via manure application events and thus increase the uncertainty in the model predictions on SOC stocks.

Our modelled regional-scale results are consistent with a recent meta-analysis finding that N-fixing legume cover crops contribute more to increasing SOC storage than do non-legume plants (Abdalla et al., 2019). However, it should be noted that in LPJ-GUESS we





assumed that cover crops in Eastern Africa are rotated with the main crops and thus solely grown during the short rainy season. This

assumption is likely to result in cover crop biomass input to the soil pools being too high as we may overestimate the length of the

fallow period for cover crop growth (Porwollik et al., 2021). Such an overestimation would then also be reflected in high SOC

estimates. At present more than 90% of total annual crop yields in Ethiopia are achieved in the long rainy season (Central Statistical

Agency, 2016); nevertheless, most farmers are reluctant to implement a "crop + cover-crop" rotation since this practice still requires

sacrificing one (short) season of maize production. Our model experiments support earlier findings that planting leguminous cover

crops during the short rainy season is expected to sustainably achieve SOC and may lead to yield increases in the tropics of SSA (Rao

and Mathuva, 2000; Carsky et al., 2001), although yield benefits from N-fixing cover crops at some smallholder farms may not fully

compensate for the production loss of the short rainy season (Carsky et al., 2001).

The view that adopting CA techniques can increase SOC storage in agricultural soils for SSA is based on analyses of differences

between management practices, but without a time perspective (Martinsen et al., 2019; Kihara et al., 2020). The future projections

done here point that the observed SOC decline in 12-year trials in western Kenyan (Figs. 2-3) would continue to be found in other

parts of Eastern Africa under most assessed management practices with the exception of $F_{conserv}$ (Fig. S7), in line with the finding of a

recent modelling study in Kenya (Nyawira et al., 2021). The 4p1000 initiative (https://www.4p1000.org/) launched at COP 21 in Paris

sets a target of 3.4 Pg C $yr^{-1}$ SOC sequestration in agricultural soils (0-40cm) worldwide to contribute to mitigating global climate

change (Corbeels et al., 2019). Our modelling results indicate that croplands situated in Eastern Africa can achieve this target only if a

combination of management practices would be adopted and sustained. But even though altered management practices may not

always support a positive soil C sequestration at regional scale (especially under climate change), they nonetheless are here projected

to lower SOC losses and have co-benefits for crop production (Kihara et al., 2020; see also Figs. 5-6).

### 4.3.2 Cropland N loss and yields

Compared to North America, Western Europe, and East Asia, annual total N loss from agricultural soils in SSA is rather small, mainly

due to the low N fertilizer use across the region (Liu et al., 2010; Bouwman et al., 2013). Our model simulated N loss of 45-134 Gg N

$yr^{-1}$ in Kenya, is approximately half of statistics-based estimates of 76-297 Gg N $yr^{-1}$, using a nitrogen-budget method (Zhang et al.,

2021). Likewise, our standard management experiment ($F_{std}$) simulated regional mean N loss of 9.2 kg N $ha^{-1}$ $yr^{-1}$ (Table 5), again

about half of the estimated 16.7-18.2 kg N $ha^{-1}$ $yr^{-1}$ in Eastern Africa reported by Kaltenegger et al. (2021). One possible reason for

these discrepancies may be missing processes in LPJ-GUESS, such as N loss via soil erosion or surface runoff. Also, the nitrogen-

budget method adopted in Zhang et al. (2021) and Kaltenegger et al. (2021) assumed that all the crop residues were retained in the

soils after harvest, contrasting with the set-ups in our $F_{std}$ simulation (only 10% of residue retention, Table 3). Removing most residues

from cropland in the model experiment is expected to produce low N loss because of less N inputs to the soils compared to 100% of

residue retention. With all the residues left in the fields ($F_{RR}$, Table 3), the model regionally showed N loss of 20.8 Gg N $yr^{-1}$ (Table 5),

comparable to the findings in Kaltenegger et al. (2021).

For untilled maize and pulses cropping systems, we found a negative correlation between simulated crop production and N loss (Fig.

S5), implying that yield increases derived from N loss reduction are possible in Eastern Africa by no-till management in the medium

to long term. However, at least in our simulations, these production benefits may not be attained in the first two decades of simulations

(Fig. 6c) because of the time that soil C and N pools need to adjust to the change in management. A similar finding emerged also from

a meta-analysis of Pittelkow et al. (2015b), which pointed to increased yields, globally, in cereals and legumes cropping systems only

after more than 10 years of conversion from a conventional tillage to a no-till system.

An increase of 89% in N losses was simulated across the study region in response to leaving crop residues in the field compared to the

model standard management (Fig.5a). A global analysis found that leaving residues behind might increase gaseous N emission by 8-



37% (Xia et al., 2018), but the same study also estimated straw return to reduce hydrological N losses by 10-26%. At present the crop residues implementation and soil representation in LPJ-GUESS do not account for soil hydraulic properties in response to residue

application. This missing process is likely to result in an overestimation of hydrological N losses since straw return generally reduces N leaching through enhancing soil water retention in reality (Blanco-Canqui et al., 2007). In addition, crop residues after harvest in Eastern Africa are expected to rapidly decompose in response to the warm and moist climate, continuously releasing reactive N for plant uptake in the subsequent cropping period (Kihara et al., 2015). As discussed earlier, only a single growing season within a year was represented in LPJ-GUESS, and without cover crops, the relatively long bare fallow period under the simulated residue retention

systems would amplify N losses as the mineralized N is not used to support plant growth during the short rainy season. Nevertheless, modelled crop production induced by retained residues still increased by 18% on regional average (Fig. 5a) because of the enhanced size of the mineral N pool. This result is in line with two previous studies that showed a mean increase of 19-35% over a three-year period in Ethiopian wheat systems with 66% residue retention (Adimassu et al., 2019), and another reporting -1-39% of maize yield changes in response to residue retention management in semi-arid Kenya (Kihara et al., 2011).

In some simulated grid cells, using cover crops moderately enhanced N losses, in particular for N-fixing cover crops ($F_{CC\text{-}BNF}$, Fig. 5a). One possible explanation for this simulation is that the enhanced available N derived from the fast decomposition of cover crops would serve as substrate for N losses instead of being taken up by the main crop, mainly due to the temporal inconsistency between periods of soil N mineralization and high N demand of the main crop (Marcillo and Miguez, 2017). Compared to the bare fallow model set-up (see $F_{std}$, Table 3), our simulations regionally predicted a slight yield increase of 6% in non-legume cover crop systems

but a high increase of 19% for N-fixing cover crops (Fig. 5a), supporting to the meta-analysis findings that legume cover crops usually contribute more to increasing subsequent crop yields than non-legumes when N fertilizer inputs are low (Tonitto et al., 2006; Quemada et al., 2013; Marcillo and Miguez, 2017; Thapa et al., 2018).These slight yield benefits (also reduced productivity in few grid cells; see Fig.5a) found in non-legume cover crop simulations primarily resulted from indirect competition for water and nutrients (Valkama et al., 2015), which may not be available for the following main crops planted in the long rainy season.

**4.4 Trade-offs and win-win management options**

In our study, we attempted to identify synergistic management strategies for achieving environmental sustainability without compromising crop production in Eastern Africa. None of the assessed management options fully achieved a win-win situation in terms of increasing soil C stocks and crop production while minimizing N losses when integrated over the study regions. Synergies and trade-offs among the three examined indicators varied between locations (Fig. 5a) and cropping systems (Fig. S5).

From the perspectives of food demand and SOC sequestration only, conservation agriculture (CA, $F_{conserv}$), as an integrated management with no-tillage, residue and manure application, and N-fixing cover crops included, was simulated to be the most promising practice for both present-day conditions and future scenarios. Nevertheless, considering the potential yield reduction in the first several years under CA systems (Stevenson et al., 2014; Pittelkow et al., 2015b), it may be difficult to convince smallholder farmers to adopt such a practice; if indeed 1-25% of crop production loss could be expected compared to the standard management

practice (Fig. 6c), farmers would suffer economic losses despite the accompanying 1-10% of increase in SOC storage (Fig. 6a). Furthermore, labor demand and cost-ineffective investment in CA maintenance may prevent this practice from being implemented widely in Eastern Africa (Thierfelder et al., 2013; Kihara et al., 2020). However, in our study this practice was modelled as the only one showing a net SOC sequestration in the future, with annual carbon uptake rates of 1.1 and 2.7 Tg C yr$^{-1}$ between 2015 and 2100 for Kenya and Ethiopia, respectively (Fig. S7). The economic considerations would be potentially quite different if, in a carbon

trading scheme, land management that leads to enhanced carbon sequestration would receive monetary compensation for the resulting yield reductions.



Rather than adopting a fully integrated CA, it is more common to use N-fixing legumes as cover crop or intercrop in smallholder farming system over Eastern Africa (Rao and Mathuva, 2000; Ngome et al., 2011). This crop management approach in our simulations ($F_{CC-BNF}$) had positive impacts also for soil C storage and food production, with low environmental cost in terms of N losses (Fig. 5). This win-win situation could be also sustained under future climate change (Fig. 6). However, it should be noted that the absence of soil pH constraints to legume inoculation in LPJ-GUESS (Ma et al., 2021) most likely results in an overestimate of the N fixation rate in the $F_{CC-BNF}$ simulation. Much experimental evidence from African farms indicates that the fixed N amount under legume-based cropping systems in SSA can be as low as 0-50 kg N ha$^{-1}$ yr$^{-1}$, primarily caused by the inconsistent effectiveness of inoculation in the acid soils (Ulzen et al., 2016; Muleta et al., 2017; Vanlauwe et al., 2019). The overestimated N fixation in the model may thus bias the contribution of legume cover crops to the C-N cycle and crop production -but possibly also to N losses. To better represent cover crop management, the evaluation of modelled N-fixing herbaceous legumes against targeted field experiments, together with implementation of multi-cropping systems (see Sect.4.3.1) are needed in future model work.

**5 Summary**

In this study we presented a large-scale modelling analysis with LPJ-GUESS, highlighting potential long-term effects of management practices on crop ecosystems in Eastern Africa under different climate change scenarios. The modelled C-N variables and crop yields in responses to varying agricultural practices were evaluated. Our results showed that crop ecosystems represented in LPJ-GUESS realistically responded to different management strategies and climate variation and produced soil C stocks, N losses, and crop productivity comparable to measurements in the studied region.

Our model demonstrated that the effects of management on agricultural ecosystems in Eastern Africa can be beneficial for climate change mitigation without compromising crop yields, in particular for the combined conservation agriculture practice with all soil C conserving techniques included. This integrated strategy was the only practice simulated to potentially achieve a positive SOC sequestration under climate change. Adopting N-fixing cover crop systems was identified as a dominant practice to regionally increase food production and C storage in agricultural soils, with low environmental costs in form of N losses. This win-win situation was shown to persist under a range of future climate pathways. However, processes missing from the model, such as multi-cropping system and N losses via runoff and soil erosion, might have biased our assessed management effects on crop ecosystems regionally.

The adoption of these management practices by farmers is promising from a climate change mitigation perspective but perhaps difficult to achieve in reality because of the yield losses in the first several years under conservation agriculture systems. Farmers are mostly risk-averse when faced with new management practices. To change this situation, a payment scheme for carbon sequestration legislated by the government or volunteered by corporations and individuals (Salzman et al., 2018) may be needed to fully compensate for farmers' economic losses in Eastern Africa, particularly in the context of future environmental change.

**Code and data availability**

Global historical climate data of GSWP3-W5E5 and future climate projection from five GCMs (ISIMIP3b) are available at https://www.isimip.org/gettingstarted/input-data-bias-correction/ (last access: 14 Nov. 2021). The monthly climate forcing data set of CRUJRA can be downloaded at https://data.ceda.ac.uk/badc/cru/data/cru_jra/ (last access: 14 Nov. 2021). National yield statistics (FAOSTAT) of six crop types presented in this paper are from http://www.fao.org/faostat/en/#data/QC (last access: 19 Aug. 2021). The code and post-processing scripts used in this study are available upon request to the corresponding author.

**Supplement**

The supplement related to this article is available online at XXX



**Author contributions**

AA and JM conceived this study. JM, SSR, ADB, PA and AA designed the experimental protocol runs. JM performed all the simulations and carried out the analysis. SSR and PA processed the model input data globally. SSN provided two long-term field trials data in western Kenya for model evaluation. JM wrote the original draft, with all authors contributing to the revisions.

**Competing interests**

The authors declare that they have no conflict of interest.

**Acknowledgements**

This research has been supported by German Federal Ministry for Economic Cooperation and Development (BMZ) and administered through the Deutsche Gesellschaft für Internationale Zusammenarbeit (GIZ) Fund for International Agricultural Research (FIA), grant number 81206681.

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
