# Peer review of "Assessing the impacts of agricultural managements on soil carbon stocks, nitrogen loss and crop production — a modelling study in Eastern Africa"

_Biogeosciences, 2021_

## Author Comment (AC1)

**Response to anonymous referee #1** (Comments in black, Answers in blue, Suggested revisions in green)

**General comments:**

*This is a comprehensive regional modelling study into the impacts of cropland management on soil C sequestration, crop yields and nitrogen loss. Ma and co-authors used a DGVM model LPJ-GUESS with implementation of C-N cycle and six representative crop types to investigate trade-offs regarding these ecosystems services under the present-day and future climate pathways in Eastern Africa.*

*I think that the study is interesting and innovative, and topic is highly relevant to Biogeosciences. This is a challenging simulation work with large uncertainties in Kenya and Ethiopia, but authors do a good job in comparison with meta-analysis from published literature and give an in-depth discussion on model limitations of each management practice. Generally, the manuscript is well structured and written with reasonable simulation design and evaluation results. My comments are minor and mostly limited to the text where are a bit difficult to follow, especially in the description of model protocol runs.*

We thank the reviewer for the expressed interest in our manuscript. In the revisions to the manuscript we will be addressing the raised questions as described below.

**Specific comments**

*L23: What is 'standard management' here? It refers to Fstd in Table 3, but it is unclear to readers before getting through the whole text, please consider to revise it.*

Yes, 'standard management' is the $F_{std}$ simulation in Table 3. To make it clear, 'standard management' will be revised to 'conventional management' throughout the manuscript as suggested, and we will give an explanation on standard simulation in the footnote of Table 3: "std – standard simulation, representing a conventional management prevalent in Eastern Africa."

*L90: How does LPJ-GUESS regionally represent crop sowing and harvest? More details on the computation of crop phenology would be appreciated.*

Similar to most ecosystem and crop models, LPJ-GUESS adopts crop-specific accumulated heat requirements to model plant growth and development. Crops are allowed to adapt to the local climate by dynamically adjusting the heat requirements (potential heat units, PHU) to different climatic zones (Lindeskog et al., 2013).

LPJ-GUESS can compute suitable sowing dates based on local climate with five seasonality types incorporated (Waha et al., 2012). The five seasonality types are determined by temperature and precipitation conditions, reflecting different intra-annual variability of temperature and precipitation. LPJ-GUESS uses specific rules per seasonality type to simulate sowing date. For example, in semi-arid regions, crops are sown at the onset of the main rainy season, which is defined as the largest sum of monthly precipitation-to-potential-evapotranspiration ratios of four consecutive months (Lindeskog et al., 2013). Crops are harvested each year when prescribed heat sum requirements are fulfilled (or maximum growing season length is reached). Multi-cropping systems within a year and intercropping are not yet implemented in LPJ-GUESS.

We will add the information on the calculation of sowing and harvest dates to Sect. 2.1 in the new manuscript. Suggested revision: "Planting date is determined dynamically based on local climatology in each grid cell with five seasonality types represented (a combination of temperature- and precipitation-limited behaviors; Waha et al., 2012), and crops are harvested once every year when accumulated heat requirements are fulfilled (Lindeskog et al., 2013). At this point multi-cropping and intercropping are not yet incorporated in the model."

*L122: What about N-fixing grass as intercrops in the model? No?*

Intercropping systems (two crops at the same time growing beside each other in a crop field) are not yet implemented in LPJ-GUESS so far, thus, N-fixing grass cannot be simulated as intercrops in the model. We will clarify the intercropping information in Sect.2.1 in the revised manuscript (see previous response).

*L127: Residue removal fraction here is set to 90%, but the proportion is 50% for site-scale evaluation in Sect 2.3.1(L179). Please clarify the setup difference between regional and site simulations.*

We will clarify that 90% of residue removal is only used for regional simulations. Suggested revision: "In this study, we increase the residue removal fraction to 90% for the regional simulations based on observations from Ethiopia."

*L176: The second maize growing period modelled as N-fixing grass makes nonsense to me because of the extra N addition to the soils from BNF.If it's the case, what is the N fixation rate of the grass? Why not using non-N-fixing cover crop as a replacement?*

We agree with the reviewer's opinion that N-fixing grass could bring additional N input to the soils. The second maize growing period was thus modelled as non-legume cover crop in our original experiment. See the description in Sect. 2.3.1: "In this study, the second growing period under the continuous maize systems was modelled as a non-N-fixing cover crop with all the above-ground biomass removed from the field at both sites."

*L178: 'assuming an N content of 1.75% in the farmyard manure'. Please provide a reference.*

Reference will be added to the end of this sentence according to the comment. Suggested revision: "assuming an N content of 1.75% in farmyard manure with a fixed C:N ratio of 16 (Gichangi et al., 2006; Nyawira et al., 2021)"

*L188: 'assumed that INM3 was under grassland systems for the period 1901-2002 '. How many above-ground biomass is removed from ecosystem here? Author stated that in the model 50% of AG biomass in C3 grass is harvested every year in Sect. 2.1 (L87), but the none of biomass is supposed to be removed here since grassland is natural vegetation without any managements.*

As the reviewer mentioned, none of grass biomass were removed from ecosystems for the period 1901-2002 in our simulation since they are natural vegetation called "grassland systems", in which human disturbance, such as harvest, is not supposed to take place. In comparison, grassland with 50% of AGB harvested every year is called "pasture systems" in the model. To make it more clear, this sentence will be revised to: "assumed that INM3 was under grassland systems for the period 1901-2002 with all the above-ground biomass returned to the soils".

*L185-192: It will be helpful to understand the impacts of LUC on soil C storage if time series of SOC over 1901-2002 are also presented somewhere apart from Figs. 2 and 3. Perhaps in the Supplement.*

We thank the reviewer for this suggestion and will add the desired figure to the SI of the revised version of the manuscript.

[Figure]

**Figure S2.** The modelled SOC stocks (0-150cm) by LPJ-GUESS at the CT1 and INM3 sites between 1901 and 2002. The shaded areas represent the period of cropland systems. All the above-ground biomass in grassland systems was returned to the soils in the simulations (see Sect. 2.3.1).

*L231: Is the 'standard' simulation representing the real-world management for smallholder farming systems in Eastern Africa?*

Yes, this is correct. We will give an explanation on standard simulation in the footnote of Table 3: "std – standard simulation, representing a conventional management prevalent in Eastern Africa."

*L240: 'standard' management between 1901-2014 is a basic simulation, providing soil C-N pools background for alternative management practices from 2015-2100. why the year of 2014 is the only time break over the whole simulated period (200 years), rather than 2006, or other years? The historical LUH2 data set ends up in 2014, but RCP climate data terminates in 2006, no?*

In this study, apart from the observation-based CRUJRA climate forcing, the model was also driven by GCMs-based SSP climate scenarios from CMIP6, in which historical forcing ends at the year 2014 and future projection starts from the year 2015 (Eyring et al., 2016). As the reviewer mentioned, the historical land use data set from LUH2 terminates in 2014 as well. Therefore, the year 2014 in our protocol runs is selected to distinguish historic period and future projection over 1901-2100.

*L246: 'All simulated outputs in the last ten years of the model experiments were taken for analysis'. You mean only the outputs from 2091-2100 taken for analysis? Maybe we shouldn't name it '2091-2100' because you are using the constant climate and CO2, this period doesn't correspond to the real calendar years like RCP scenarios.*

Yes, 'the last ten years' refers to the period of 2091-2100. We agree with the reviewer's perspective that '2091-2100' in B2 simulation is different from the real calendar years. To avoid confusing readers, we described it as 'the last ten years of the model experiments' in the manuscript.

*L255-259: Authors are trying to quantify the potential transition of Fopt caused by future climate change through comparing the difference between C2 and C3, but B2 is much representative for the present-day climate, why not B2 vs. C3? any Fopt difference in spatial pattern between B2 and C2?*

The observation-based CRUJRA data and GCMs-based (bias-corrected, see Table 1) climate are used as model inputs in B2 and C2 simulations, respectively, both representing the climate condition of the historic period. These two data sets, however, differ to some degree (see Fig.S4c below) since in C2, the output was simulated by GCMs. To avoid any climate inconsistency between the historic and future periods, the GCMs-based C3 simulation (future) thus needs to compare with GCM-based C2 (historic), so that climate patterns are consistent when computing the relative change (future vs. historic).

[Figure]

**Figure S4.** Inputs used for simulations in Eastern Africa: (a) simulated crop-specific areas (rain-fed and irrigated) over the historic period compared to statistics from FAO (ha); (b) mean rates of mineral N fertilizer and manure applied to each crop type (kg N ha$^{-1}$) from 1901-2014; (c) annual $CO_2$ concentration (ppm, right scale), and mean temperature (°C, left scale) from CRUJRA (B1, Table 1) and five GCMs (C1 and C2, Table 1). The faba bean and common bean are simulated as pulses in (a) and (b). The black, blue, green, and red thick solid lines in (c) denote the temperature averaged by five GCMs for historical, SSP1-RCP26, 3-70 and 5-85 scenarios, respectively; thin lines in light color represent the temperature from individual GCM. Dashed lines show $CO_2$ concentrations.

We compared the $F_{opt}$ spatial difference between B2 and C2 simulations under the present-day climate (see figure below), finding that a general agreement of $F_{opt}$ spatial pattern exists between two simulations, despite some deviations for individual grid cells.

[Figure]

The modelled optimal soil C sequestration management ($F_{opt}$, Eq.4) under the present-day climate in Kenya and Ethiopia driven by (a) the observation-based CRUJRA data set (B2, Table1) and (b) the GCMs-based climate (C2, Table1). The SOC outputs in the last ten years of the two simulations were taken for analysis.

*Figs. 2-3: Please also add the simulated SOC results between 1901 and 2002 at two sites.*

See previous response, Figure S2 will be added to the SI in the revised manuscript.

*Figs.1-3 and Table 4: Please consider to change SOC unit from 't C ha-1' to 'Mg C ha-1', the latter one is more common in soil carbon studies.*

SOC unit is revised to 'Mg C ha$^{-1}$' in the main text and figures as suggested.

*Fig. 4a and 4c: There are sharp decreases of reported crop production in 2007-2008 in Kenya, which are not captured by the model. what caused this? climate input used for simulations?*

There are two potential factors resulting in maize reduction in Kenya over 2007-2009: (1) Before the 2007 domestic presidential elections, Kenya had been hit by months of drought (Huho and Mugalavai, 2010), and (2) the post-election violence in 2008 further worsened the situation. The conflicts left potentially productive farmland unattended. For (1), we are unable to evaluate if the climate data set used for simulations match the observed precipitation as no observed records are available for comparison regionally. For (2), Gitau and Meyer (2019) estimated that approximately 20% of the maize crop went unharvested as a result of political unrest that led to many farmers displaced in 2008. LPJ-GUESS cannot represent societal or political impacts on crop production.

*L361: 'Leaving most parts of crop residues in the field'. It would be good if revising 'most part of ' to 'all', since in Frr simulation in Table 3 the fraction of residue retention is 100%.*

"most part of crop residue" is corrected to 'all the crop residue' based on suggestion.

*L381: Look at the crop production in Fig.4, sorghum and pulse yields are largely overestimated, indicating that the updating growth parameters described in Sect. 2.3.2 does not work well, I expected to see some explanations on such deviations.*

Wortmann et al. (2009) reported that insect pests, particularly shoot flies and stalk borers, have been identified as a major constraint to sweet sorghum production in SSA due to the high sugar content in the crop, resulting in yield reduction of ca. 15-88% in Eastern Africa. LPJ-GUESS does not yet take pests into account, which could contribute to the large overestimation of

sorghum production. Regarding pulses, Ma et al. (2022) argued that the high legume N fixation capacity modelled by LPJ-GUESS in warm and moist climates would result in overestimation of grain legume production, mainly because BNF may reduce the N constraints on leaf photosynthesis and subsequently strengthen the flow of carbon assimilation to storage organ.

We will give an in-depth discussion in Sect. 4.2 to explain the potential reasons: "A strong overestimation in pulses production was seen for both countries (Fig. 4). This can be likely explained by the high legume N fixation capacity modelled by LPJ-GUESS in warm and moist climates (Ma et al., 2022). A high rate of BNF may reduce the N constraints on leaf photosynthesis and subsequently strengthen the flow of carbon assimilation to storage organs, resulting in high production in N-fixing crops. Yet, similar to pulses, our simulated sorghum yields at country level were also significantly greater than FAO records (Fig. 4). This suggest that other factors are at play as well. For example, insect pests, particularly shoot flies and stalk borers, have been identified as the major constraint to sorghum production in SSA (Wortmann et al., 2009), leading to an estimated yield reduction of 11-49% in western Africa and 15-88% in Eastern Africa (Okosun et al., 2021). LPJ-GUESS does not yet take pests into account, which could contribute to the large overestimation of sorghum production in our studied region. Additionally, a good representation of photosynthate allocation to various plant organs is important when modelling crop yields (Bondeau et al., 2007). In this study we updated the daily assimilate partitioning scheme of sorghum based on the existing literature (Fig. S3), but this process has not yet been parameterized and calibrated against observations from field experiments. Whether or not this related to the large-scale yield overestimation needs to be further investigated in future work."

*Table 5: It seems that the modelled total SOC stocks, N losses and crop productions in both countries show a fairly good agreement with previous studies. The issue here is if the simulated crop-specific areas are also comparable with other statistics? For instance, the cropland area in 2014 is 6,222,100 and 17,433,400 ha for Kenya and Ethiopia, respectively (L398). How do we know these numbers are reliable?*

We compared the simulated crop-specific areas (rain-fed and irrigated) over the historic period with statistical harvested areas from FAO (see Fig. S4a above), finding that the modelled areas generally agree with FAO-based records in maize, pulses, sorghum and wheat, which are the top four widely grown crops in Kenya and Ethiopia. Fig. S4 will be added to the SI in the revised manuscript.

*L441: If 50% of residue retention in the model setup is not fully equivalent to the observed input of 2 t ha-1, how many residues in absolute value was left in the field in simulations?*

The quantity of 50% of residue retained depends on individual years and managements. Generally, around 1.3-2.2 and 1.4-2.4 t ha$^{-1}$ yr$^{-1}$ of maize residue across all years and managements in our simulations are left in the fields in CT1 and INM3 sites, respectively (see figure below).

We will add the relevant information of maize residue quantity in Sect. 4.1 in the revised manuscript: "In addition, compared to the fixed amount of maize residue retained in the field trials (2 t ha$^{-1}$), using 50% of residue retention in the model set-up will introduce some variation in terms of C inputs to soils because of the varying biomass of simulated maize residue between years (~1.3-2.2 and 1.4-2.4 t ha$^{-1}$ of residue returned to the fields in CT1 and INM3 simulations, respectively; not shown)."

[Figure]

The modelled 50% of maize residue retained to the soils across all managements at INM3 (a) and CT1(b) sites between 2003 and 2016 (t ha$^{-1}$ yr$^{-1}$). N0, N30, N60, and N90 represent the treatments that maize receives N fertilizer rate of 0, 30, 60, and 90 kg N ha$^{-1}$, respectively. See Table 2 for the details of each treatment.

*L447: Land use history prior to experiments is most likely the reason to explain the declined SOC at two Kenyan sites. Look at Fig.6, soil C pools take almost 20-30 years to reach a new equilibrium after shifting the managements. The 10+ years of cultivation at the evaluated sites is thus not long enough for the stable C-N pools.*

We agree with the reviewer's ideas that both land use history and management duration contribute to SOC changes. They are discussed in Sect. 4.1: "All investigated management practices led to declining SOC stocks in the field trials (Figs. 2 and 3), the overall trends were also reproduced by the model. Nonetheless, the soil C losses rates from 2004-2015 were unexpected, since addition of farmyard manure and residues can enhance SOC storage via additional C inputs to soils while conservation tillage slows down decomposition in the SOM pools. Both INM3 and CT1 sites in this study were under natural grassland before the trials started (see A1 and A2, Table 1); hence, SOC losses in observations and simulations reflected a) grassland soils tending to store more carbon than cropland, and b) a new SOC equilibrium may not have been reached in the maize cropping systems after 10+ years of cultivation (Lal, 2008). A similar finding was reported by Moebius-Clune et al. (2011), who showed declining SOC in western Kenya even after more than 50 years of conversion from primary forest to maize. Furthermore, fast turnover of the SOM in the humid tropics could be another factor affecting the SOC trends because of the prevailing warm and moist climate (Olin et al., 2015a). The turnover-driven C losses at the sites may exceed the gains resulting from the C addition from manure and residue application (Kihara et al., 2020; Nyawira et al., 2021)."

*L452: 'because of the prevailing warm and moist climate ', but most part of Kenya belongs to semi-arid climate to my mind, no?*

Yes, most part of Kenya are arid or semi-arid climate. However, the two experiment sites discussed here in western Kenya are characterized by warm and moist climate (Nyawira et al., 2021). We will point out that it refers to western part of the country. Suggested revision: "Furthermore, fast turnover of the SOM in the humid tropics could be another factor affecting the SOC trends because of the prevailing warm and moist climate (i.e., western Kenya in this study)."

*L480: what about soybean yields? also overestimated in the model? Soybean is not main cash crops in Kenya and Ethiopia, wondering if model shows the similar performance in N-fixing crop productions in these two countries caused by overestimated BNF rates.*

Similar to pulses, the model also tended to overestimate soybean yield on national level in these two countries because of the high N fixation rate simulated by LPJ-GUESS. More details on soybean evaluation can be found in Ma et al. (2022).

*L493: 'vary between 15.5-32.7 Tg yr-1'. Please correct it to '15.9-32.7 Tg yr-1'. In Zomer et al (2017), SOC increase rate under medium scenario is 13.22 (Ethiopia)+2.7(Kenya)=15.9 Tg yr-1, 27.17(Ethiopia)+5.5 (Kenya)=32.7 Tg yr-1 under high scenario.*

Thanks for this comment, the range will be corrected to '15.9-32.7 Tg yr$^{-1}$' in the revised manuscript.

*L533: 'crop+cover-crop rotation'. Do you mean 'main crop (long rainy season)+ cover crop (short rainy season)' or ''main crop 1 (long rainy season)+ main crop 2 (short rainy season)+cover crop (dry season)' ?*

Since double-cropping systems within a year haven't been implemented to LPJ-GUESS (see previous response), only a single growing season of the main crop is modelled in this study. Thus, 'crop+cover-crop rotation' here refers to 'main crop (long rainy season) + cover crop (short rainy season)'. To make it more clear to readers, this sentence will be revised to: "most farmers are reluctant to implement a 'main crop (long rainy season) + cover-crop (short rainy season)' rotation system since this practice still requires sacrificing one (short) season of maize production."

*L538-547: Good discussion on SOC storage from the time perspective, apart from different management practices comparison. It's interesting to see that only combined Fconserv run show the net CO2 uptake in the future, while other strategies present stable or declined SOC (Fig.S7). Any possibility to implement the simulation of Integrated Soil Fertility Management (ISFM) here? which is the most well-known soil conserving techniques in SSA (Sommer et al., 2018).*

Thanks for this positive comment. We do agree with the reviewer in that ISFM is the most well-known soil conserving strategy in Eastern Africa. In fact, manure application and mineral N fertilizer, as the two most important components of ISFM management, have been taken into account in our simulations (see Table 3).

*L567-570: What if you combine the hydrological N losses and gaseous N emission in Xia et al (2018)? will simulated total N losses be consistent with the meta-analysis?*

Xia et al. (2018) reported that straw return at a global scale increased total N losses due to a greater stimulation of gaseous N emissions than the reduction in N leaching and runoff, which is also supported by our simulations in Eastern Africa. However, we are not able to straightforwardly combine hydrological N losses with gaseous N emission to get a value of total N losses in Xia et al. (2018), since the original data are not provided in their study.

*L610-615: 'the fixed N amount under legume-based cropping systems in SSA can be as low as 0-50 kg N ha-1 yr-1'. It's unclear to me, N fixation rate of 0-50 kg N ha-1 yr-1 refers to legume crops or N-fixing grass? What is the BNF rate of cover crop in FCC-BNF simulation? I'm asking because as we know, the N benefit to soil fertility from green manure is literally correlated with N fixation capacity.*

It mainly refers to the leguminous plants, including grain and forage legumes (such as soybean, faba bean, and N-fixing grasses). Our modelled N fixed by herbaceous legumes in the F$_{CC-BNF}$ simulation are generally around 10-40 kg N ha$^{-1}$ yr$^{-1}$, with the highest value of 70-90 kg N ha$^{-1}$ yr$^{-1}$ found in southern Ethiopia and western Kenya (see Fig.S9). Figure S9 will be added to the SI in the revised manuscript.

To make it more clear, the sentence will be revised as: "For example, modelled N fixed by herbaceous legumes can be up to 70-90 kg N ha$^{-1}$ yr$^{-1}$ in some grid cells (Fig. S9), while experimental evidence from African farms indicates that the nodulation of roots in grain and forage legumes in SSA may not be successful, primarily due to the inhibition of inoculation in the acid soils (Ulzen et al., 2016; Muleta et al., 2017; Vanlauwe et al., 2019)".

[Figure]

**Figure S9.** The N fixation rate (kg N ha⁻¹ yr⁻¹) of herbaceous legumes modelled by LPJ-GUESS in the $F_{CC-BNF}$ simulation under the present-day climate, averaged over the last ten years of simulation (B2, Table 1).

**References**

Bondeau, A., Smith, P. C., Zaehle, S., Schaphoff, S., Lucht, W., Cramer, W., Gerten, D., Lotze-campen, H., Müller, C., Reichstein, M. and Smith, B.: Modelling the role of agriculture for the 20th century global terrestrial carbon balance, Glob. Chang. Biol., 13(3), 679–706, doi:10.1111/j.1365-2486.2006.01305.x, 2007.

Eyring, V., Bony, S., Meehl, G. A., Senior, C. A., Stevens, B., Stouffer, R. J. and Taylor, K. E.: Overview of the Coupled Model Intercomparison Project Phase 6 (CMIP6) experimental design and organization, Geosci. Model Dev., 9(5), 1937–1958, doi:10.5194/gmd-9-1937-2016, 2016.

Gichangi, E. M., Karanja, N. K. and Wood, C. W.: Composting cattle manure from zero grazing system with agro-organic wastes to minimise nitrogen losses in smallholder farms in kenya, Trop. Subtrop. Agroecosystems, 6(1870–0462), 57–64, 2006.

Gitau, R. and Meyer, F.: Spatial price transmission under different policy regimes: A case of maize markets in Kenya, African J. Agric. Resour. Econ., 14(1), 15–27, 2019.

Huho, J. M. and Mugalavai, E. M.: The Effects of Droughts on Food Security in Kenya, Int. J. Clim. Chang. Impacts Responses, 2(2), 61–72, doi:10.18848/1835-7156/cgp/v02i02/37312, 2010.

Kihara, J., Bolo, P., Kinyua, M., Nyawira, S. S. and Sommer, R.: Soil health and ecosystem services: Lessons from sub-Sahara Africa (SSA), Geoderma, 370(July 2019), 114342, doi:10.1016/j.geoderma.2020.114342, 2020.

Lal, R.: Soil carbon stocks under present and future climate with specific reference to European ecoregions, Nutr. Cycl. Agroecosystems, 81(2), 113–127, doi:10.1007/s10705-007-9147-x, 2008.

Lindeskog, M., Arneth, A., Bondeau, A., Waha, K., Seaquist, J., Olin, S. and Smith, B.: Implications of accounting for land use in simulations of ecosystem carbon cycling in Africa, Earth Syst. Dyn., 4(2), 385–407, doi:10.5194/esd-4-385-2013, 2013.

Ma, J., Olin, S., Anthoni, P., Rabin, S. S., Bayer, A. D., Nyawira, S. S. and Arneth, A.: Modeling symbiotic biological nitrogen fixation in grain legumes globally with LPJ-GUESS ( v4.0 , r10285 ), Geosci. Model Dev., 15(2), 815–839, 2022.

Moebius-Clune, B. N., van Es, H. M., Idowu, O. J., Schindelbeck, R. R., Kimetu, J. M., Ngoze, S., Lehmann, J. and Kinyangi, J. M.: Long-term soil quality degradation along a cultivation chronosequence in western Kenya, Agric. Ecosyst. Environ., 141(1–2), 86–99, doi:10.1016/j.agee.2011.02.018, 2011.

Muleta, D., Ryder, M. H. and Denton, M. D.: The potential for rhizobial inoculation to increase soybean grain yields on acid soils in Ethiopia, Soil Sci. Plant Nutr., 63(5), 441–451, doi:10.1080/00380768.2017.1370961, 2017.

Nyawira, S. S., Hartman, M. D., Nguyen, T. H., Margenot, A. J., Kihara, J., Paul, B. K., Williams, S., Bolo, P. and Sommer, R.: Simulating soil organic carbon in maize-based systems under improved agronomic management in Western Kenya, Soil Tillage Res., 211(March), 105000, doi:10.1016/j.still.2021.105000, 2021.

Okosun, O. O., Allen, K. C., Glover, J. P. and Reddy, G. V. P.: Biology, Ecology, and Management of Key Sorghum Insect Pests, J. Integr. Pest Manag., 12(1), doi:10.1093/jipm/pmaa027, 2021.

Olin, S., Lindeskog, M., Pugh, T. A. M., Schurgers, G., Wärlind, D., Mishurov, M., Zaehle, S., Stocker, B. D., Smith, B. and Arneth, A.: Soil carbon management in large-scale Earth system modelling: Implications for crop yields and nitrogen leaching, Earth Syst. Dyn., 6(2), 745–768, doi:10.5194/esd-6-745-2015, 2015.

Ulzen, J., Abaidoo, R. C., Mensah, N. E., Masso, C. and AbdelGadir, A. A. H.: Bradyrhizobium inoculants enhance grain yields of soybean and cowpea in Northern Ghana, Front. Plant Sci., 7, 1–9, doi:10.3389/fpls.2016.01770, 2016.

Vanlauwe, B., Hungria, M., Kanampiu, F. and Giller, K. E.: The role of legumes in the sustainable intensification of African smallholder agriculture: Lessons learnt and challenges for the future, Agric. Ecosyst. Environ., 284(December), 106583, doi:10.1016/j.agee.2019.106583, 2019.

Waha, K., Van Bussel, L. G. J., Müller, C. and Bondeau, A.: Climate-driven simulation of global crop sowing dates, Glob. Ecol. Biogeogr., 21(2), 247–259, doi:10.1111/j.1466-8238.2011.00678.x, 2012.

Wortmann, C., Mamo, M., Mburu, C., Letayo, E., Birru, G. A., Kaizzi, K. C., Chisi, M., Mativavarira, M., Xerinda, S. and Ndacyayisenga, T.: Atlas of Sorghum Production in Eastern and Southern Africa, Lincoln, United States., 2009.

Xia, L., Lam, S. K., Wolf, B., Kiese, R., Chen, D. and Butterbach-Bahl, K.: Trade-offs between soil carbon sequestration and reactive nitrogen losses under straw return in global agroecosystems, Glob. Chang. Biol., 24(12), 5919–5932, doi:10.1111/gcb.14466, 2018.

---

## Author Comment (AC2)

**Response to anonymous referee #2** (Comments in black, Answers in blue, Suggested revisions in green)

**General comments:**

*The LPJ-GUESS model was used here to examine seven crop management practices and their effect on soil carbon (C) pool, nitrogen (N) loss, and crop yields under different climate scenarios that is the present-day climate situation in Eastern Africa and its potential for the future. The study tackles an important topic and will allow us to improve our understanding of how improved agricultural management can protect soils and lessen soil greenhouse gas emissions in Eastern Africa, where there are currently very few such data available. I thus believe that the topic is very interesting and of great relevance to Biogeosciences. In terms of design and evaluation results, the manuscript is well written with a good structure. The authors have really done their work in the discussion of the results which are well referenced. I believe the work is very relevant and very important, apart from a few very minor adjustments that should be made to the manuscript.*

We thank the reviewer for the expressed interest in our manuscript. In the revisions to the manuscript we will be addressing the raised questions as described below.

*For example, the authors should briefly explain what they mean by standard management and standard simulation in the manuscript, which has been used throughout the manuscript.*

Thanks for this comment. We will revise the 'standard management' to 'conventional management' throughout the manuscript as suggested, and give an explanation on standard simulation in the foot note of Table 3: "std – standard simulation, representing a conventional management prevalent in Eastern Africa."

*In Kenya, beans and maize are intercropped primarily. Since beans represent one of Kenya's major crops, is there any reason they weren't included here? Otherwise, this work for me has been well done.*

We do agree with the reviewer's comment that common beans and faba bean are very important crops in Kenya and Ethiopia, respectively. Currently, crops in LPJ-GUESS are modelled as crop functional types (CFTs, see Table S1), i.e. group of crops with similar functional behaviors. In this study, beans with N fixation capacity are simulated as pulses, which in the model generally stands for faba bean, common bean, cowpea, etc. To make it clear, we will add an explanation in Sect. 2.3.2: "In this study we performed simulations with six CFTs一maize, pulses (representing faba bean and common bean), sorghum, wheat, rice, and soybean一which are grown widely in Kenya and Ethiopia."

We also compared the simulated pulses areas (rain-fed and irrigated) over the historic period with statistical harvested areas from FAO (see Fig. S4a below), finding that the modelled areas generally agree with FAO-based records in two countries and thus that our simulation does reflect the changes of pulses planting areas in the past several decades. Fig. S4 will be added to the SI in the revised manuscript. Meanwhile, we will highlight that common bean and faba bean in this study are simulated as pulses in the caption of Fig.S4:"Inputs used for simulations in Eastern Africa: (a) simulated crop-specific areas (rain-fed and irrigated) over the historic period compared to statistics from FAO (ha); (b) mean rates of mineral N fertilizer and manure applied to each crop type (kg N ha$^{-1}$) from 1901-2014; (c) annual $CO_2$ concentration (ppm, right scale), and mean temperature (°C, left scale) from CRUJRA (B1, Table 1) and five GCMs (C1 and C2, Table 1). The faba bean and common bean are simulated as pulses in (a) and (b). The black, blue, green, and red thick solid lines in (c) denote the temperature averaged by five GCMs for historical, SSP1-26, 3-70 and 5-85 scenarios, respectively; thin lines in light color represent the temperature from individual GCM. Dashed lines show $CO_2$ concentrations."

[Figure]

**Figure S4.** Inputs used for simulations in Eastern Africa: (a) simulated crop-specific areas (rain-fed and irrigated) over the historic period compared to statistics from FAO (ha); (b) mean rates of mineral N fertilizer and manure applied to each crop type (kg N ha⁻¹) from 1901-2014; (c) annual $CO_2$ concentration (ppm, right scale), and mean temperature (°C, left scale) from CRUJRA (B1, Table 1) and five GCMs (C1 and C2, Table 1). The faba bean and common bean are simulated as pulses in (a) and (b). The black, blue, green, and red thick solid lines in (c) denote the temperature averaged by five GCMs for historical, SSP1-26, 3-70 and 5-85 scenarios, respectively; thin lines in light color represent the temperature from individual GCM. Dashed lines show $CO_2$ concentrations.

**Specific comments**

*LN 380 Is there an explanation for the overestimation in production in pulses and sorghum in Fig 4.*

Wortmann et al. (2009) reported that insect pests, particularly shoot flies and stalk borers, have been identified as a major constraint to sweet sorghum production in SSA due to the high sugar content in the crop, resulting in yield reduction of ca. 15-88% in Eastern Africa. LPJ-GUESS does not yet take pests into account, which could contribute to the large overestimation of sorghum production. Regarding pulses, Ma et al. (2022) argued that the high legume N fixation capacity modelled by LPJ-GUESS in warm and moist climates would result in overestimation of grain legume production, mainly because BNF may reduce the N constraints on leaf photosynthesis and subsequently strengthen the flow of carbon assimilation to storage organ.

   Since both referees (see the comments from Referee#1) are concerned about the reasons for the overestimation of crop yields in the model, we will give an in-depth discussion in Sect. 4.2 to explain the potential reasons: "A strong overestimation in pulses production was seen for both countries (Fig. 4). This can be likely explained by the high legume N fixation capacity modelled by LPJ-GUESS in warm and moist climates (Ma et al., 2022). A high rate of BNF may reduce the N constraints on leaf photosynthesis and subsequently strengthen the flow of carbon assimilation to storage organs, resulting in high production in N-fixing crops. Yet, similar to pulses, our simulated sorghum yields at country level were also significantly greater than FAO

records (Fig. 4). This suggest that other factors are at play as well. For example, insect pests, particularly shoot flies and stalk borers, have been identified as the major constraint to sorghum production in SSA (Wortmann et al., 2009), leading to an estimated yield reduction of 11-49% in western Africa and 15-88% in Eastern Africa (Okosun et al., 2021). LPJ-GUESS does not yet take pests into account, which could contribute to the large overestimation of sorghum production in our studied region. Additionally, a good representation of photosynthate allocation to various plant organs is important when modelling crop yields (Bondeau et al., 2007). In this study we updated the daily assimilate partitioning scheme of sorghum based on the existing literature (Fig. S3), but this process has not yet been parameterized and calibrated against observations from field experiments. Whether or not this related to the large-scale yield overestimation needs to be further investigated in future work.”

*LN447: I wonder if you can explain the unexpectedly low soil C sequestration rates from 2004-to 2015 by looking at the history of the experimental field.*

We agree with the reviewer's idea that land use history would affect SOC sequestration. We add the discussion in Sect. 4.1: " Both INM3 and CT1 sites in this study were under natural grassland before the trials start (see A1 and A2, Table 1), hence SOC losses in observations and simulations reflected a) grassland soils tending to store more carbon than cropland, and b) a new SOC equilibrium may not have been reached in the maize cropping systems after 10+ years of cultivation (Lal, 2008).” In addition, we also add the plot of the modelled SOC over 1901-2002 (Fig S2) to the SI in the revised manuscript, where the readers can clearly see the simulated impacts of land use change on SOC stocks at both sites.

[Figure]

**Figure S2.** The modelled SOC stocks (0-150cm) by LPJ-GUESS at the CT1 and INM3 sites between 1901 and 2002. The shaded areas represent the period of cropland systems. All the above-ground biomass in grassland systems was returned to the soils in the simulations (see Sect. 2.3.1).

*LN452. In Kenya, 80% of the land is semi-arid and arid. I think you need to point out the western part of the country, however.*

Thanks for this comment, this sentence will be revised to: “Furthermore, fast turnover of the SOM in the humid tropics could be another factor affecting the SOC trends because of the prevailing warm and moist climate (i.e., western Kenya in this study).”

*LN 458 Did you observe any termite mounds in the experimental fields?*

Yes, termite activity was observed in the field experiments. Kihara et al. (2015) studied the impacts of termite activity on SOC at the same sites that we used for evaluation.

**References**

Bondeau, A., Smith, P. C., Zaehle, S., Schaphoff, S., Lucht, W., Cramer, W., Gerten, D., Lotze-campen, H., Müller, C., Reichstein,

M. and Smith, B.: Modelling the role of agriculture for the 20th century global terrestrial carbon balance, Glob. Chang. Biol., 13(3), 679–706, doi:10.1111/j.1365-2486.2006.01305.x, 2007.

Kihara, J., Martius, C. and Bationo, A.: Crop residue disappearance and macrofauna activity in sub-humid western Kenya, Nutr. Cycl. Agroecosystems, 102(1), 101–111, doi:10.1007/s10705-014-9649-2, 2015.

Lal, R.: Soil carbon stocks under present and future climate with specific reference to European ecoregions, Nutr. Cycl. Agroecosystems, 81(2), 113–127, doi:10.1007/s10705-007-9147-x, 2008.

Ma, J., Olin, S., Anthoni, P., Rabin, S. S., Bayer, A. D., Nyawira, S. S. and Arneth, A.: Modeling symbiotic biological nitrogen fixation in grain legumes globally with LPJ-GUESS ( v4.0 , r10285 ), Geosci. Model Dev., 15(2), 815–839, 2022.

Okosun, O. O., Allen, K. C., Glover, J. P. and Reddy, G. V. P.: Biology, Ecology, and Management of Key Sorghum Insect Pests, J. Integr. Pest Manag., 12(1), doi:10.1093/jipm/pmaa027, 2021.

Wortmann, C., Mamo, M., Mburu, C., Letayo, E., Birru, G. A., Kaizzi, K. C., Chisi, M., Mativavarira, M., Xerinda, S. and Ndacyayisenga, T.: Atlas of Sorghum Production in Eastern and Southern Africa, Lincoln, United States., 2009.